

# Evaluation of drought representation and propagation in Regional Climate Model simulations over Spain

Anaïs Barella-Ortiz[1] and Pere Quintana-Seguí[1]

[1]Observatori de l'Ebre (Universitat Ramon Llull - CSIC), Roquetes (Spain)

**Correspondence:** : Anaïs Barella-Ortiz (abarella@obsebre.es)

**Abstract.** Drought is an important climatic risk, of complex modeling due to the interaction of different processes and their corresponding temporal scales. It is expected to increase in intensity, frequency and duration due to a warmer climate. Therefore, it is vital to know the evolution of drought in relation to climate change. For this, the better understanding of processes involved in it is key. The study here presented, analyses drought representation and propagation by regional climate models from Med-CORDEX simulations by means of standardized indices. The models used are RCSM4, CCLM4, and PROMES. Focus is made on three types of drought: meteorological, soil moisture and hydrological. Results show that these models improve meteorological drought representation, but large uncertainties are identified in its propagation and the way soil moisture and hydrological droughts are characterised. These are mainly due to the relevance of model formulation. For instance, it affects the temporal scale at which precipitation variability propagates to soil moisture and streamflow. In addition, downscaling is also seen to affect streamflow variability, and thus hydrological drought.

*Copyright statement.* TEXT

## 1 Introduction

According to the Intergovernmental Panel on Climate Change (IPCC) Fifth Assessment Report, impacts from recent climate-related extremes show a significant vulnerability and exposure of ecosystems and human systems to current climate variability (IPCC, 2014). These affect countries with different development level, in both urban and rural areas, and are directly related to a lack of both prediction and preparation regarding this phenomena. What is more, most of these impacts have been seen to increase in a climate change scenario. For instance, drought frequency has increased over Mediterranean regions during the last decades (Mariotti, 2010; Sousa et al., 2011).

Drought is one of the climate-related extremes. It is a complex phenomenon with important impacts on the environment (Reichstein et al., 2013; Vicente-Serrano et al., 2013), as well as on the society and economy (FAO, 2009; Owens et al., 2003). Drought is normally explained as a long deficit of water caused by a lack of precipitation. However, the concept of drought is wider, being this definition limited to meteorological drought. There are several types of drought (IPCC, 2014), depending on which part of the system suffers from water deficit:





- Meteorological drought: is explained in the previous paragraph. According to the IPCC, it corresponds to the period where precipitation has been considerably lower than its average level.

- Soil moisture drought (also known as agricultural drought): is due to a deficit of water in the soil's unsaturated zone, making impossible to cover the crop's needs. It should be stressed that there are no thresholds for this drought, since it depends on the type of crop, its location, and season.

- Hydrological drought: occurs when streamflows and water levels in rivers, lakes, and groundwater are low. A more precise definition refers to superficial and groundwater water availability decrease in a managing system, affecting water demand.

- Environmental drought: is a combination of the previous types.

Since the precipitation deficit propagates through the hydrological system (Wilhite, 2000), drought types are related between them. The soil moisture drought appears slightly out of phase of the meteorological drought in non irrigated areas and depends on several factors, like the soil's type and capacity to retain water, as well as potential evapotranspiration. However, it is related to the availability of irrigation water in irrigated areas, thus depending on the hydrological drought. The hydrological drought is also affected by the meteorological drought, but at a larger temporal scale than the soil moisture drought. It can appear months, even some year (mainly in the case of groundwater) after a precipitation deficit. However, if there is no further delay in precipitation, it can occur that no hydrological drought is observed. Therefore, each system component is characterized by its own propagation dynamics and memory.

The IPCC informs about little confidence in drought trends observed at a global scale due to i) lack of direct observations, ii) dependency on the drought definition used, and iii) geographical unconsistencies (IPCC, 2014). However, it does see probable that its frequency increases in arid zones at the end of the 21st Century.

Drought is a difficult phenomenon to model, since complex interactions between atmospheric and continental surface processes have to be combined, together with human action (Van Loon, 2015). In addition, relations between drought types and the issues explained previously, add complexity to its modelling. However, modelling is an interesting tool to analyse processes related to drought, which also permits to perform prediction and impact studies using different scenarios (Van Loon et al., 2012).

Land Surface Models (LSMs) describe physical processes in the soil-vegetation-atmosphere interface, simulating water and energy budgets. Since they are physical models, they take into account most of the processes involved in drought and its propagation (Vidal et al., 2010b). LSMs can be used coupled to climatic and meteorologic models or uncoupled (offline), forced by reticular databases of atmospheric variables. In the latter case, they act as a physical distributed hydrological model, which can be used to study droughts. In addition, this phenomenon representation is further improved by adding a routing scheme that permits the simulation of streamflow and water volumes using runoff. An example is the *SIM (SAFRAN-ISBA-MODCOU)* model (Habets et al., 2008). Other studies following this line are: Balsamo et al. (2012); Decharme et al. (2012); Martínez de la Torre (2014).



Besides LSMs, there are also Regional Climate Models (RCMs), which provide a series of aspects that add value to climate modelling. One of their greatest advantages is their adaptation to a regional scale (Feser et al., 2011). This allows to use regional observations and to increase the model's physical parametrization complexity. In addition, using a higher resolution permits these models to perform studies of atmospheric phenomena at a small scale. Finally, a RCM simulation is cheaper than

a General Circulation Model (GCM) one in terms of computing resources.

Drought studies are normally performed using drought indices, among which the Standard Precipitation Index (SPI) (Mc-Kee et al., 1993) and the Palmer Drought Severity Index (PDSI) (Palmer, 1965) are some of the most widely used. These studies have gained interest in the last years due to the increasing concern regarding i) complexity in hydrological resources management, ii) rise in frequency, severity, and duration of extreme events, iii) increase in both, vulnerability and exposition

to suffer it, and iv) climate change (Andreadis and Lettenmaier , 2006; Mishra and Singh, 2010, 2011; López-Bustins et al., 2013; Jenkins and Warren, 2014). These issues make evident the need to work on how droughts will behave in the future. With respect to modelling, a better understanding of the different types and their propagation processes is key to improve drought representation. This will enable the development of new tools and the improvement of the existing ones regarding prediction and management.

The work here presented, analyses how RCMs simulate drought. For this, we evaluate the RCMs capacity to represent and propagate drought over mainland Spain, by comparing differences between models and how relevant they are. It allows to analyse the RCM's contribution with respect to the global model that drives it, and see whether if it improves drought simulation.

PaiMazumder et al. (2013) and Masud et al. (2017) analyse projected changes to drought aspects in Canadian prairies by

means of RCMs. The first study uses three severity drought indices and an ensemble of Canadian RCM simulations. The second one uses the SPI and PDSI indices and an RCM ensemble too. Maule et al. (2012) use SPI and a version of PDSI to analyse drought representation by 14 RCMs from the ENSEMBLES project (van der Linden and Mitchell, 2009) at a European scale. Blenkinshop and Fowler (2007a, b) analyse drought characteristics (like duration and severity) using RCMs from the PRUDENCE project (Christensen et al., 2007) over Britain and at a European scale by means of the Drought Severity Index

(DSI). Other studies use RCMs to analyse drought trends (Wu et al., 2016) and study the capacity of downscaling data to represent drought spells (Anagnostopoulou, 2017). Even though, studies are focused on meteorological drought, there are others like Vu et al. (2015) and Meresa et al. (2016) that analyse hydro-meteorological drought by computing precipitation and runoff standardized indices using RCM simulations. Soil moisture and hydrological droughts are also analysed according to the time accumulated period considered to compute a given drought index. For example, according to García-Valdecasas

Ojeda et al. (2017), a 3 month time scale will provide information about agricultural drought spells, while a 12 month time scale will do so for hydrological drought spells. Other studies following this line are Vicente-Serrano and Lóopez-Moreno (2005),Vicente-Serrano (2006), and Edossa et al. (2009).

The RCM simulations used in this study are downloaded from the Med-CORDEX database (Ruti et al., 2015). All of them use ERA-Interim (Balsamo et al., 2012; Dee et al., 2011) as their driving data. However, models using different surface

schemes are selected in order to analyse to what degree they differ when representing drought. These are: RSCM4 (Nabat et al.,



2014), CCLM4 (Rockel et al., 2008), and PROMES (Domínguez et al., 2010). As reference data, the SAFRAN atmospheric analysis (Durand et al., 1993, 1999), LSM offline simulations (from the ISBA-3L (Noilhan and Planton, 1989; Noilhan and Mahfouf, 1996) and ORCHIDEE (De Rosnay and Polcher, 1998; Krinner et al., 2005) LSMs) and a simulation from the SIMPA hydrological model (Estrela and Quintas, 1996; Ruiz, 1999) are used.

The work focuses on three types of drought, meteorological, soil moisture, and hydrological. Each of them is characterized using standardized indices, which define drought according to the variability of the studied variable. For instance, the SPI is computed with precipitation and used to study meteorological drought. According to Farahmand and AghaKouchak (2015), standardized indices can be computed using other variables. Therefore, the SSMI, computed with soil moisture, is used to analyse soil moisture drought, and the SSI and SRI, computed with streamflow and total runoff, respectively, are used to

analyze hydrological drought.

In the next section we will detail the study area. Next, Sect. 3 provides a description of the forcings and driving data, as well as the different models used in this study. Section 4 informs about the methodology used. Sections 5 and 6, present the results and discussion. Finally, conclusions and perspectives are given in Sect. 7.

## 2   Area and study period

This study is carried out over mainland Spain, where the main climatic regimes are oceanic and mediterranean. The time period has been limited to the availability of RCM simulations and observations: from 1989 to 2008 for the meteorological and soil moisture drought, and from 1989 to 2005 for the hydrological drought.

The first column of Fig. 1 shows the mean annual precipitation over the study period provided by ERA-Interim and the RCMs (SAFRAN's precipitation is also shown in panel b). It is larger over the north and north-west (exceeding $2000 \ \mathrm{mmy}^{-1}$)

as well as over mountainous chains, indicating the strong influence of relief. It is lower along the main basins' valleys due to the orographic shadow effect, and is minimum over the south-east, which is the driest region of the peninsula. Over this area, precipitation does not exceed $100 \ \mathrm{mmy}^{-1}$ in some regions, resulting in a semiarid climate or even a desert-like one (Aemet, 2011). We would like to stress SAFRAN's significant contrast in the relief due to its higher resolution and the use of data from the dense pluviometric network of the Spanish State Meteorological Agency (AEMET). This evidences the peninsula's

complex spatial structure of precipitation.

Spain is a semi-arid region and it is not densely vegetated. Soil moisture displays a large annual cycle. In fact, soil moisture anomalies in Spring may influence droughts and heat waves in Europe (Vautard et al., 2007; Zampieri et al., 2009).

From a hydrological point of view, there is a strong dependence of the main rivers on the precipitation generated in the nearby relief and the resulting runoff. An example is the Ebro basin and the Pyrenees. In addition, the impact of the anthropic effect

must be taken into account, since there is a wide network of dams and river canals which, in some cases, operates between basins. It was constructed to deal with some issues like the distance between the resource and the demand location or the need to store water to use during dry periods.



According to Sousa et al. (2011), drought in Spain has increased in severity and frequency. With respect to meteorological drought, precipitation does not show significant annual trends. However, a reduction in Spring and Summer has been observed (de Luis et al., 2010), as well as an increase in the number of consecutive dry days (Turco and Llasat, 2011). Both aspects have an impact on soil moisture drought, which is also affected by a rise in annual and seasonal temperatures (DelRio2011 et al.,

2011; Kenawy et al., 2013). It increases the atmospheric demand (Vicente-Serrano et al., 2014), and thus evapotranspiration, reducing the soil's water content. As for the hydrological drought, we must also consider the advance in the thaw date and a thinning of the blanket of snow in mountainous areas, like the Pyrenees (Morán-Tejeda et al., 2013), which affect streamflows and increase this type of drought.

## 3   Datasets

This section describes the products used to force LSMs and to drive RCM simulations Sect.3.1. It also depicts the models used in this study: RCMs, LSMs, and the SIMPA hydrometeorological model, Sect.3.2.

### 3.1   Forcings and driving data

Table 1 provides information about forcings and the driving data used in this study.

### 3.1.1   SAFRAN meteorological analysis

SAFRAN (Système d'Analyse Fournissant des Renseignements Atmosphériques à la Neige) is a meteorological analysis system (Durand et al., 1993, 1999) developed by Météo-France.

It provides accurate estimates of the following variables: precipitation, 2 m temperature, 10 m wind speed, 2 m relative humidity, and cloudiness, as well as modelled downward visible and infrared radiation following the radiation scheme of (Ritter and Geleyn, 1992). For this, an optimal interpolation algorithm (Gandin, 1966) that combines observations and a first guess is

used. The first guess employed is ERA-Interim for all variables, except for precipitation, that is obtained from observations. The meteorological station data belong to the Spanish Meteorological State Agency (Agencia Estatal de METeorología, AEMET) network.

Precipitation is analysed by means of daily observations, while the rest of variables are analyzed every 6 hours. The data is then hourly interpolated, using different methods that depend on the variable.

Although SAFRAN was originally created as an atmospheric forcing analysis in mountainous regions for avalanche forecasting, its use has been extended to other matters following operational and research purposes. This is shown in the works of Quintana-Seguí et al. (2008) and Vidal et al. (2010a) in France, or Szczypta et al. (2015) in Morocco. As for the case of Spain, SAFRAN was extended for a 35 year period (1979/1980 – 2013/2014), detailed in Quintana-Seguí et al. (2016). In addition, it was implemented and validated over the Ebro basin (north-eastern Spain) (Quintana-Seguí et al., 2017).

In this study the standard version of SAFRAN, at a 5 km resolution has been regridded to a 30 km resolution, to compare it with the RCM simulations in the meteorological and soil moisture drought analyses. From hereafter, it will be referred to as



**Table 1.** Forcings / driving data

| | Product | Description | Spatial resolution (km) | Temporal resolution (hour) | Reference |
|---|---|---|---|---|---|
| SAFRAN | SFR | Analysis | 5 | 1 | (Quintana-Seguí et al., 2017), |
| | SLR (Low Resolution) | | 30 | | (Quintana-Seguí and Barella-Ortiz, in review) |
| ERA-Interim | ERA | Reanalysis | 80 | 3 - 6 | (Dee et al., 2011) |

SLR (SAFRAN Low Resolution). However, the standard version is used for the hydrological drought analysis. From hereafter, it will be referred to as SFR.

### 3.1.2 ERA-Interim

The ECMWF Interim Re-Analysis (ERA-Interim) provides a global atmospheric reanalysis. In this study, the ERA-Interim
product (Balsamo et al., 2012; Dee et al., 2011), which starts in 1979 and is continuously updated in real time, is used.

It is produced by means of a data assimilation system based on a 2006 release of ECMWF's Integrated Forecast System, IFS (Cy31r2) and uses a 4-dimensional variational analysis (4D-Var) with a 12 hour analysis window. The database has atmospheric and surface parameters with a temporal scale of 6 and 3 hours, respectively. The spatial resolution is 80 km with 60 vertical levels, from the surface to 0.1 hPa. It improves some important issues of ERA-40, like the hydrological cycle's representation.

ERA-Interim is a well-known atmospheric forcing used in a large number of studies. For instance, Belo-Pereira et al. (2011) and Quintana-Seguí et al. (2017) have validated it over the Iberian Peninsula. However, ERA-Interim is known to have biases that affect land-surface simulations in a negative way. According to Ngo-Duc et al. (2005) and Weedon et al. (2011), they can be corrected.

From hereafter, it will be referred to as ERA.

## 3.2 Models

Table 2 provides information about the models (RCMs, LSMs, and hydrometeorological) used in this study.

### 3.2.1 Land Surface Models

LSMs are bidirectionally coupled to the RCMs' atmospheric model in order to simulate surface processes. They can be run coupled to another model (i.e. climatic, atmospheric, etc.) or in an uncoupled mode (offline). In the latter case, models are
forced by reticular databases. So the atmospheric feedback is not taken into account. This makes them a good tool to study drought, since LSMs can be handled like physical distributed hydrological models.



Since there is no observed truth for soil moisture available, LSM offline simulations are used in this study to analyse soil moisture drought.

### 3.2.2 SURFEX

SURFEX (SURFace EXternalisée) is Météo-France's surface modelling platform. It consists of several independent physical
models for natural land surface (ISBA), urbanized areas (TEB), and water bodies (FLake) and oceans, among others. In addition, SURFEX can simulate surface processes regarding chemistry and aerosols and be used to assimilate surface and near surface variables. It can be coupled to atmospheric and hydrological models or run in a stand-alone mode.

The ISBA (Interaction Sol-Biosphère-Atmosphère) LSM (Noilhan and Planton, 1989; Noilhan and Mahfouf, 1996), developed by the CNRM (Centre National de Recherches Météorologiques), is composed by various modules which simulate heat
and water transfers in the soil, vegetation, snow, and surface hydrology. This scheme has evolved using different approaches to model the soil. As a result, there are several versions that can be used. For this study, we have selected the ISBA-3L (Boone et al., 1999), which considers a three layered description of the soil. It should be stressed that it is limited by certain aspects. For example, underground water is not represented and there are no horizontal water transfers.

SURFEX uses the ECOCLIMAP 1 km resolution, land cover database (Masson et al., 2003; Kaptue et al., 2010; Faroux et
al., 2013), which considers more than 550 cover types all over the world. Its vegetation variability depends on the location, climate, and phenology. Most of ISBA's soil and vegetation parameters are derived from the ECOCLIMAP database. More precisely, from the cover types and soil texture.

In this study, two offline SURFEX simulations, using the 3 layer ISBA version and forced with i) SAFRAN and ii) ERA-Interim, are used as references to study soil moisture and hydrological droughts. From hereafter this LSM will be referred to
as ISB.

### 3.2.3 ORCHIDEE

The ORCHIDEE (Organising Carbon and Hydrology In Dynamic EcosystEms) LSM (De Rosnay and Polcher, 1998; Krinner et al., 2005) was developed by the "Institut Pierre-Simon Laplace" (IPSL). It can be run in a stand-alone mode or coupled to the "Laboratoire de Météorologie Dynamique" (LMD-Z) General Circulation Model (GCM) (Li, 1999), developed by the LMD in
Paris.

Hydrology is approached by means of a diffusive equation with a multilayer scheme. For this, the Fokker-Planck equation is solved considering a soil depth of 2 m distributed in eleven layers. The fine resolution is key to better model the interaction between the root profile and the soil moisture distribution at different depths, as well as infiltration processes. In addition, ORCHIDEE includes a sub-grid variability of soil moisture. For this, each grid box is divided into three soil moisture profiles
with different vegetation distribution, but the same soil texture and structure, obtained from the Zobler map (Post and Zobler, 2000).

In this study, an offline ORCHIDEE simulation, forced with ERA, is used as reference to study soil moisture and hydrological droughts. From hereafter this LSM will be referred to as ORC.



### 3.2.4 Hydrological models

SIMPA is the Spanish acronym meaning "Integrated System for Rainfall-Runoff Modelling" (Sistema Integrado de Mod-elización Precipitación-Aportación) (Estrela and Quintas, 1996; Ruiz, 1999). It is a conceptual distributed hydrological model for water management, developed by the Spanish centre for studies and experimentation on public works ("Centro de Estudios

y Experimentación de Obras Públicas", CEDEX).

SIMPA provides estimates of the water cycle's main components, such as precipitation, evapotranspiration, and river dis-charge at a monthly scale in natural regime. This regime is characterized by the free flowing of water, with no aspect unrelated to the environment (like dams for water resources management) affecting it.

Data is provided on a $1 \, \mathrm{km}^2$ grid. The model employs its own forcing, HIDRO (Quintas, 1996), which uses an observational
dataset similar to that of SAFRAN.

In this study SIMPA is used as a reference product for streamflow, to analyse the hydrological drought. From hereafter, it will be referred to as SMP.

### 3.3  Regional climate models

Med-CORDEX (Ruti et al., 2015) is a contribution to the Coordinated Regional Climate Downscaling Experiment (CORDEX,
Giorgi et al. (2009)), which focuses on the Mediterranean region. Its objective is to improve the reliability of the past and future information through the study of Mediterranean climate variability and trends. For this, regional atmospheric, land surface, river and oceanic climate models and regional climate system models are available for the research community.

In this study, drought representation and propagation of three RCMs is analysed. For this, three RCM simulations from the Med-CORDEX database are selected: RCSM4, CCLM4, PROMES. Each one of them uses a different surface scheme and,
therefore, represents in different ways physical processes related to precipitation, soil moisture, and surface and sub-surface runoff. All the RCM simulations are driven with ERA, Sec. 3.1.2.

CNRM-RCSM4 (Sevault et al., 2014; Nabat et al., 2014) is a RCM developed by the "Centre National de Recherche Météorologique" (CNRM). It includes the regional climatic atmospheric model ALADIN-Climate (Radu et al., 2008; Déqué and Somot, 2008; Farda et al., 2010; Colin et al., 2010; Herrmann et al., 2011), the ISBA LSM, the TRIP routing scheme
(Decharme et al., 2010), and the regional ocean model NEMOMED8 (Beuvier et al., 2010). From hereafter, it will be referred to as RS4.

The COSMO-CLM (CCLM) model (Rockel et al., 2008) is the climate version of the COSMO model, developed by the Goethe Universität Frankfurt (GUF). The surface scheme is a multilayer version of the Jacobsen and Heise (1982) two layer model. From hereafter, it will be referred to as CL4.
The PROMES model (Castro et al., 1993; Sánchez et al., 2004; Domínguez et al., 2010) is developed by the Universidad Complutense de Madrid (UCM) and the Universidad de Castilla-La Mancha (UCLM). It is coupled to the ORCHIDEE LSM. From hereafter, it will be referred to as PMS.



**Table 2.** Models used in the study

| Model | | Type | Drought | Driving data/ Forcing | Surface scheme |
|---|---|---|---|---|---|
| RCSM4 | RS4 | RCM | Meteorological, Soil moisture, Hydrological | ERA | ISBA-3L |
| CCLM4 | CL4 | | | | Multilayer |
| PROMES | PMS | | | | ORCHIDEE |
| ISBA-3L | ISB | LSM | Soil moisture, Hydrological | SAFRAN, ERA | |
| ORCHIDEE | ORC | | | ERA | |
| SIMPA | SMP | Hydrological | Hydrological | | |

### 3.4 Observations

Daily streamflow observations belong to the Spanish Ministry for the Ecological Transition (MITECO) which provides data for basins comprising more than one region. To obtain monthly series as complete as possible, only stations with at least 95% of data in the study period are selected. Certainly, the south of the peninsula is not as represented as the north, but series with

5 a larger number of data are prioritized to having a higher number of stations. From hereafter, they will be referred to as OBS.

### 4 Methodology

The methodology employed in this study is the same one used in Quintana-Seguí and Barella-Ortiz (in review). In this way, both studies complement each other, providing a wider analysis of drought representation by both LSMs and RCMs.

Drought can be characterized in several ways. Here we use standardized indices, which define drought according to the

10 variability of a given variable. For the meteorological and soil moisture drought analyses, we compute SPI and SSMI with precipitation and soil moisture, respectively. For the hydrological drought analysis we consider two indices, SSI and SRI, which are computed using streamflow and total runoff, respectively.

### 4.1 Drought indices calculation

We follow the spirit of the Standardized Precipitation Index (SPI), which allows to study meteorological drought through the

15 analysis of precipitation. Apart from the SPI itself, we compute SSMI using soil moisture, SSI using streamflow, and SRI using total runoff, to study the soil moisture and hydrological droughts. The reason for using total runoff to analyze the hydrological drought is that RCMs do not provide modelled streamflow. In fact, other studies use this variable to analyse hydrological drought (Vu et al., 2015; Meresa et al., 2016). Since we have observed (OBS) and modelled (SMP) streamflow, we decided to include them in the hydrological drought analysis by computing SSI and comparing it to SRI.





In this type of indices, the variable's time series is transformed from its original distribution to a normal one. The resulting values correspond to the number of standard deviations by which the anomaly deviates from the mean. Biases are not computed, as they are zero by construction. The computation is carried out using monthly data, and in the meteorological case, a time series of the accumulated precipitation for the previous $n$ months (being $n$ the index scale). It should be noted that in the SPI methodology, data is fitted to its corresponding parametric distribution, which can be an issue regarding studies that standardize several variables, as it is our case. To solve it, we use a non-parametric methodology, defined by Farahmand and AghaKouchak (2015).

On the one hand, meteorological drought representation by RCMs will be addressed by comparing their SPI-12 time series with those of ERA and SLR. Special attention will be payed to differences in duration, severity and area. In addition, temporal correlation of the RCMs' SPI-12 with that of ERA and SLR will be computed to identify regions where drought representation is improved, if any. On the other hand, the analysis of RCMs' soil moisture and hydrological drought representation will be performed through the calculation of the Root Mean square Differences (RMSD) and the Pearson correlation coefficient ($r$). It should be noted that an RMSD equal or greater than 0.5 often implies a change in drought category (i.e. from moderate to severe, for example), according to the SPI drought characterization scale. To facilitate the Pearson correlation analysis interpretation, the guideline proposed by Evans (1996) will be followed.

### 4.2 Meteorological drought propagation

This analysis is based on the methodology exposed by Barker et al. (2015). Soil moisture's memory to precedent precipitation will vary from one point to another, depending on its location. The reason being, that soil moisture is controlled by different aspects, like precipitation and potential evapotranspiration (climate), soil texture (soil properties), or stomatal resistance and root depth (vegetation properties), among others. In order to determine the month scale at which the precipitation deficit propagates to a soil moisture one, we carry out the following analysis:

- Compute the standardized soil moisture index with a time accumulation of 1 month (SSMI-1).

- Compute the SPI with a time accumulation of 1 to 28 months (SPI-$n_x$).

- Identify the $n_x$ scale that maximizes the correlation between SPI-$n$ and SSMI-1.

The same methodology is employed to analyse how meteorological drought propagates to hydrological drought. For this, SSI-1 is computed, instead of SSMI-1 and the SPI computation is performed using the areal mean of the basin precipitation defined by each studied gauging station.

### 4.3 Streamflow validation

The streamflow validation has been carried out using the Kling-Gupta Efficiency (KGE) (Gupta et al., 2009). The optimal value of KGE is 1, while negative values are sign of a model's bad performance.





## 5   Results

First, meteorological drought representation by RCMs is studied (Sect. 5.1). Next, soil moisture drought representation by RCMs is analysed, as well as how models address the transition from meteorological to soil moisture drought (Sect. 5.2). Finally, hydrological drought representation by RCMs and the propagation of meteorological to hydrological drought are
analysed (Sect. 5.3).

### 5.1   Meteorological drought

In this section, we will focus on precipitation because its absence is the main cause of meteorological drought. The SPI index is computed for the RCMs, ERA (driving data) and SLR (reference dataset). On the one hand, the comparison of the RCMs' SPI with that of ERA allows to identify and determine to what extent these models reproduce the structures of their driving data.
On the other hand, the comparison between the RCMs' SPI and SLR's shows to what extent RCMs improve these structures, since the SLR dataset is based on observations.

#### 5.1.1   Mean annual precipitation comparison

The first row of Fig. 1 shows mean annual precipitation of ERA and SLR, as well as their difference. The rest of rows show the RCMs' mean annual precipitation and their difference with respect to ERA (second column) and SLR (third column).

All products show greater precipitation in the north-western and northern regions of the peninsula and lower in the south-east. We would like to stress SLR's significant contrast in the relief due to its use of data from AEMET's dense pluviometric network. This evidences the complex spatial structure of precipitation in Spain. As explained in Sect. 2, relief is a determining factor in distribution. Precipitation increases in the mountain ranges and decreases in the valleys due to the effect of the shading caused by the relief. The RCMs' mean precipitation spatial structures show similar behaviour as those from ERA and SLR.

Regarding RCMs, RS4 and PMS show greater similarity between them, as well as a higher contrast than CL4. When compared to ERA, both RCMs have higher precipitation, especially in mountainous areas. This reflects the addition of water in the form of precipitation, improving the RCMs' spatial distribution of the precipitation with respect to the driving data. However, when RS4 and PMS are compared to SLR, precipitation is underestimated over some areas (valleys and coastline). It must be borne in mind that SAFRAN is mainly based on rain gauge information. There are few pluviometers in mountainous areas,
probably causing an underestimation of precipitation over these regions. However, this effect is limited to mountainous areas. The CL4 model is a different matter, since it underestimates precipitation for almost all Spain when compared to ERA and SLR.

#### 5.1.2   SPI comparison

Once the spatial distribution of precipitation in the peninsula is exposed, we study the variability of the different products and,
thus, their capacity to reproduce drought spells.





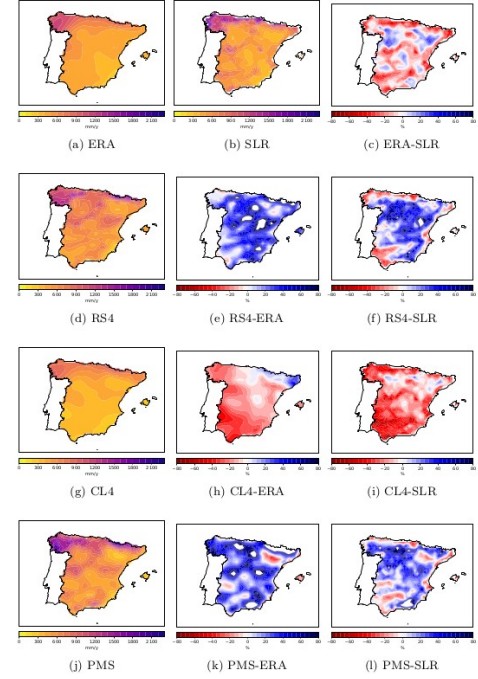

**Figure 1.** Mean annual precipitation over the study area between 1989 and 2008. Upper row: ERA and SLR mean precipitation and the difference between them. Rest of the rows: RCMs' mean precipitation and their difference with that of ERA and SLR.

Figure 2 shows the time series of SPI-12 calculated using mainland Spain's average precipitation as reproduced by ERA (panel a), SLR (panel b) and the RCMs (panels d, g, and j). The computation is performed for a time accumulation of 12 months. ERA and SLR show several drought spells occurred during the 20 years that make up the study period, being the most severe in 2005-2006. These spells coincide with those found by (Belo-Pereira et al., 2011) and appear in the RCMs' SPI time

series plots too. Therefore, RCMs are capable of reproducing these spells. However, differences in duration and severity can be observed. For instance, CL4 represents the 1992 spell with a larger duration than ERA and SLR. It is interesting to note that RCMs provide a severity of the 2002 spell which is in better agreement with that of SLR than with their driving data. In a previous study, a spurious trend in ERA was detected (Quintana-Seguí and Barella-Ortiz, in review). It can be observed in Fig. 2c, where the difference between the SPI-12 time series of ERA and SLR is represented. However, the differences between the

RCMs and SLR SPI-12 time series show that these models do not drag this trend.

To deepen the analysis of the meteorological drought's spatial structures, monthly SPI-12 maps are computed. The comparison of these structures as shown by RCMs, with those of their driving data and the reference dataset, provides more information about drought representation by RCMs. For instance, the temporal evolution of the spatial correlation of the SPI-12 maps of RCMs with ERA and SLR (not shown), indicates the similarity between RCMs with their driving data and reality (as approx-

imated by SLR), respectively. In the first case, RS4 resembles more to ERA. Despite not showing the highest correlations, it has less variability and, therefore, is more robust. The correlation of CL4 and PMS with ERA is more variable, reaching values



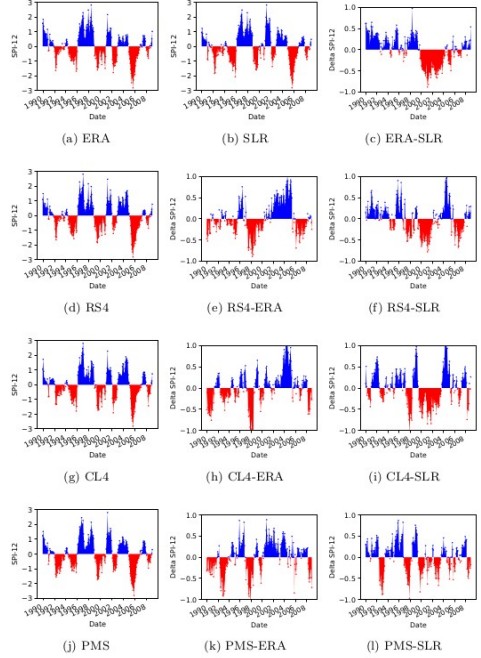

**Figure 2.** SPI-12 time series. First row: SPI-12 calculated with the spatially aggregated time series of mainland Spain precipitation, as reproduced by ERA and SLR, together with the difference between them. Rest of rows: SPI-12 time series calculated with the spatially aggregated time series of mainland Spain precipitation, as reproduced by the RCMs, together with their difference with respect to ERA (second column) and SLR (third column).

close to zero in some months, where the spatial structure of drought is not captured. In the second case, RCMs are found to correlate worse with SLR tan with ERA, as expected. It should be stressed that RS4 also shows better correlations than the other RCMs when these are compare to SLR. So, out of the three RCMs studied, RS4 deviates less from their driving data and resembles more to the reference. The correlation between ERA and SLR, shows some variability in its temporal evolution,

5    especially since 2000. This is probably due to the effect of the spurious trend identified in Fig. 2. To complement the spatial structures' analysis, Fig. 3 provides information about the difference in the percentage of area affected by drought (SPI-12 < -1) between the RCMs and i) ERA (panel a ) and ii) SLR (panel b). Differences are generally under 25%, being slightly larger those with respect to SLR. In general, RCMs show a similar behavior during a drought spell, meaning that they all overestimate or underestimate the affected area. For example, all three underestimate the area in 1994, overestimate it in 1995

10   and underestimate it in 1996. The difference is the degree to which they do it. On the one hand, in 1995, RS4 overestimates it by just over 20 %, CL4 around 15 % and PMS does not reach 10 %. On the other hand, in 1996, PMS underestimates it by more than 30 %, while RS4 and CL4 do it by 10 %. Another example is how the RCMs overestimate the area in 2000 and how they underestimate it between 2001 and 2003. RS4 is the RCM that shows the lowest percentage difference, which is coherent with our previous results.





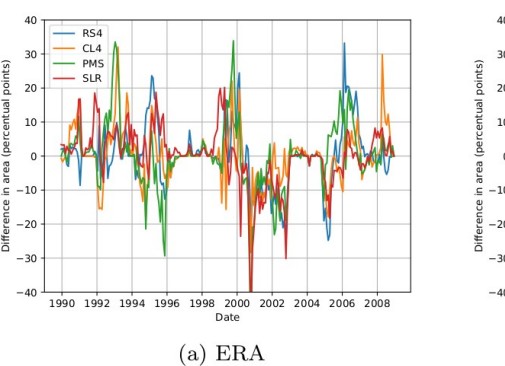
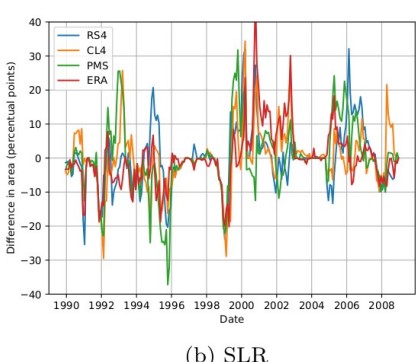

(a) ERA                (b) SLR

**Figure 3.** Differences in the proportion of area under drought (SPI-12 $< -1$) estimated by the RCMs and SLR / ERA, with respect to ERA (panel a) / SLR (panel b).

Finally, Fig. 4 provides an idea of the spatial structures of drought representation. For this, the temporal correlation (Pearson) of the SPI-12 between the RCMs and i) ERA (upper row) and ii) SLR (lower row) is computed for each grid point. Correlations of RCMs with ERA are above 0.8 in the south and worsen towards the north and the eastern coast, reaching 0.2. The Ebro basin is the region showing the poorest values, and thus, the larger differences in drought representation between RCMs and ERA. Differences in this area are also identified in the map showing the correlation between ERA and SLR, where values are even negative. However, when RCMs are compared to SLR (lower row), correlations are higher over this basin. Consequently, RCMs improve drought representation over this region.

## 5.2 Soil moisture drought

In this section, the analysis focuses on models (Table 2). More precisely, on how they reproduce soil moisture drought (Sect. 5.2.1) and its propagation from precipitation to soil moisture (Sect. 5.2.2). Offline LSM simulations from ISBA and OR-CHIDEE are used as reference. We should bear in mind that in these simulations soil processes do not impact the atmosphere, while the RCM simulations are performed in coupled mode, and thus, interact with the atmosphere.



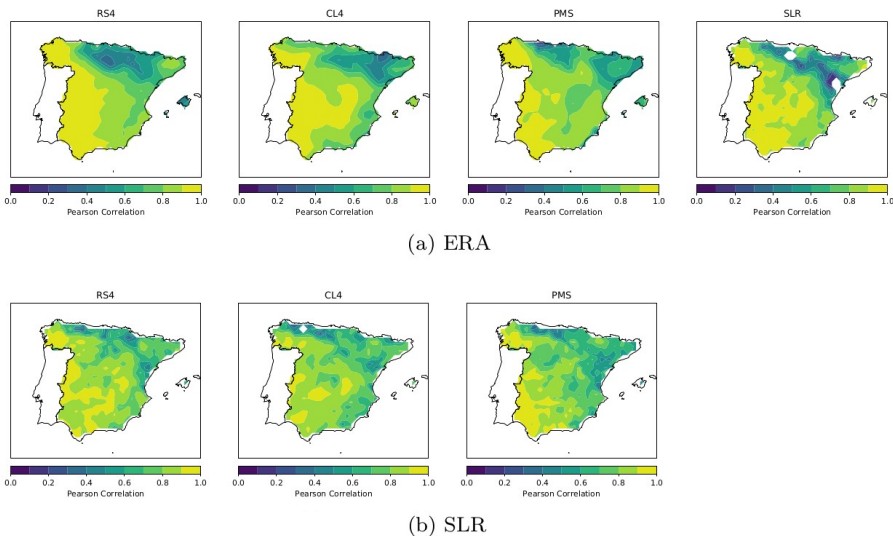

**Figure 4.** Map of the temporal correlation (Pearson) between the SPI-12 time series produced by the RCMs and SLR / ERA, with respect to ERA (upper row) / SLR (lower row).

### 5.2.1 SSMI comparison

RMSD and the Pearson correlation coefficient ($r$) are calculated comparing the SSMI from the RCM simulations and the LSM simulations. All mesh points and time steps are included in the comparison. Therefore, the results provide information regarding spatial and temporal drought structures. It should be noted that biases are not calculated because they are zero by construction (the mean of SSMI is zero). Results are shown in Table 3, the table above / below corresponds to the RMSD / $r$ calculation. A colour scale has been included to facilitate reading (green indicates a greater similarity between models than red).

To put in context these results, we will consider the drought classification according to the SPI, which is divided into 8 categories: from "extremely wet" (SPI = 2) to "extreme drought" (SPI = -2). A RMSD equal to 1 is a standard deviation of the index studied (in this case SSMI). In the framework of drought analysis, a value higher than 0.5 would imply a change of category (for example, from "slightly wet" to "moderately dry"). So, Table 3, shows that there is a change in drought category when comparing RCMs with the reference offline LSM simulations. In addition, RCMs compared between them also show a change in drought category, as expected.

Going into detail, we can observe some similarity between the RS4 simulation, which uses the ISBA surface scheme, and the ISB simulations. Compared to ISB, RS4 reproduces drought better than the other RCMs used in this study. However, this





**Table 3.** Comparison of the SSMI data from the RCM and LSM simulations. The upper block shows the RMSD and the lower one the Pearson correlation ($r$).

| RMSD | ERA-CL4 | ERA-PMS | ERA-ORC | ERA-ISB | SLR-ISB |
|---|---|---|---|---|---|
| ERA-RS4 | 1.3 | 1.26 | 1 | 0.79 | 0.81 |
| ERA-CL4 | | 0.96 | 1.12 | 1.2 | 1.24 |
| ERA-PMS | | | 1.23 | 1.17 | 1.19 |
| ERA-ORC | | | | 0.87 | 0.97 |
| ERA-ISB | | | | | 0.58 |

| $r$ | ERA-CL4 | ERA-PMS | ERA-ORC | ERA-ISB | SLR-ISB |
|---|---|---|---|---|---|
| ERA-RS4 | 0.3 | 0.36 | 0.46 | 0.66 | 0.65 |
| ERA-CL4 | | 0.65 | 0.5 | 0.35 | 0.3 |
| ERA-PMS | | | 0.39 | 0.39 | 0.37 |
| ERA-ORC | | | | 0.58 | 0.49 |
| ERA-ISB | | | | | 0.82 |

does not occur when the PMS and ORC simulations are compared, despite using the same surface scheme. In fact, RS4 and CL4 are, in general, more similar to ORC than PMS, according to the statistics. It is interesting to see the results related to the SSMI comparison of both ISB simulations. Although they show a very strong correlation, they provide an RMSD of 0.58, which could imply a change in drought category explained by the forcing used. The SSMI comparison of RCMs with both

ISB simulations (fifth column vs. sixth column) shows very similar statistics. Discrepancies could be explained by the RCMs' land-atmosphere coupling and forcing effects.

### 5.2.2 Propagation to soil moisture drought

The analysis of how meteorological drought propagates to soil moisture drought is key in hydrological resources management. In addition, it allows to detect similarities and differences in the way models address the physical processes that drive this

propagation.

  $n_x$ maps from Fig. 5 indicate the scale in months at which the correlation between SSMI-1 and SPI-$n$ is maximum, and thus, the temporal scale at which meteorological drought propagates to soil moisture drought. The first row shows RCM maps, while the second row shows ISB and ORC LSM maps. The scale goes from 0 to 28 months, being the dynamics of the model in regions with a yellowish tone slower than in regions with a bluish tone. Unfortunately, we do not have observations to compare

these results, so we will compare the models among themselves, identifying and evaluating their differences.





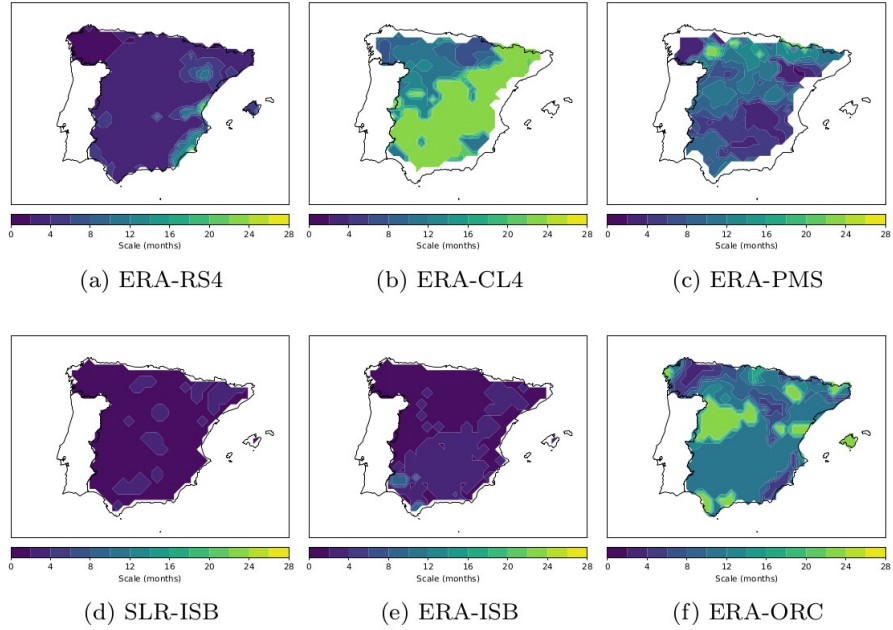

(a) ERA-RS4      (b) ERA-CL4      (c) ERA-PMS

(d) SLR-ISB      (e) ERA-ISB      (f) ERA-ORC

**Figure 5.** $n_x$ scale maximizing the correlation between SPI-$n_x$ and SSMI-1 for the RCM and LSM simulations.

The first row shows that RCMs provide different results even though they use the same driving data, which indicates the predominance of model formulation with respect to the driving data. This becomes more evident, when an RCM is compared to the LSM with the surface scheme it is coupled to. For example, RS4 and ISB maps show similar spatial patterns. These are very homogeneous, with scales that range from 1 to 4 months, implying that ISB reacts very quickly to precipitation. Another
example is the comparison between PMS and ORC. Both models show greater heterogeneity than ISB, with scales from 1 to 20 (PMS) and 24 (ORC) months, highlighting the role of the continental surface. Finally, the CL4 behaviour is quite flat. The peninsula is divided into two areas, one over the north-west, where the $n_x$ scale ranges between 6 and 12 months and a larger one with a fixed value of 20 months. It is interesting to note the similar spatial structures from the ERA-CL4 and ERA-ISB maps, indicating that soil moisture drought propagation by CL4 drags the same spatial structures as its driving data.

**5.3 Hydrological drought**

In this section we will focus on the analysis of hydrological drought. Streamflow would be the variable ideally suited for this study. However, the Med-CORDEX database does not provide simulated streamflow for none of the three RCMs. In the absence of data, it was decided to use modelled total runoff (from now on referred to as runoff) corresponding to the sub-basins defined by a selection of gauging stations. We believe that this approximation is valid because we use a coarse time step, being
larger the time propagation than the flow propagation. It should be noted that observed (OBS) and modelled (SMP) streamflow are included in the comparison.





To be able to compare streamflow with aggregated total runoff, only stations with at least 95% of observed daily data during the study period and with an area greater than 10000 $km^2$ are considered. The first criterion is set to assure that the OBS monthly series have few gaps, as explained in Sect. 3.4. The second one is defined taking into account that streamflow is approximated by runoff, which is likely to perform poorly in small basins considering the coarse resolution of the RCM simulations. It should be noted that Spanish basins are highly influenced by human management, specially those among the sizes of the selected ones. This complicates our study, since RCMs do not simulate water management procedures, but a natural regime behaviour. To single out stations that are close to this type of behaviour, we compute the KGE between SMP and OBS, setting 0.5 as the minimum threshold to consider a near-natural regime. It is an arbitrary value, which in our opinion, provides a reasonable performance with respect to SMP. We are aware that it does not represent a completely natural regime. However, it allows a fairer comparison between RCMs and observations. The final selection can be seen in Table 4, which contains the stations' code, area, and the basin they belong to.

In Sect. 5.3.1, streamflow and aggregated runoff will be compared. Next, the SSI and SRI will be compared (Sect. 5.3.2). Finally, we will analyse the way in which RCMs reproduce meteorological drought propagation to hydrological drought (Sect. 5.3.3).

### 5.3.1 Streamflow and aggregated runoff comparison

Table 4 shows the KGE comparing RCMs and LSMs aggregated runoff with SMP (upper table) and OBS (lower table), as well as the comparison between SMP and OBS. All of them are performed at a monthly scale. Green cells indicate a better performance than red ones. In addition, negative values, showing a bad performance, are marked in red.

ERA-RS4 compared to ERA-ISB and ERA-PMS compared to ERA-ORC, show that RCMs provide better KGE values than LSMs with the same surface scheme. This improvement is due to downscaling processes. However, it should be noted that RCMs show KGE values that are negative or close to zero in many stations. CL4 behaves poorly, since KGE are below 0.5. Even though RS4 is the RCM showing the highest KGE value (0.7 at station 9002), PMS behaves better over both basins. It is interesting to see the forcing effect on the ISB LSM simulations (sixth and seventh columns). When ISB is forced with ERA, the aggregated runoff performance is poorer than when it is forced by SFR, as expected.

An analysis comparing the temporal series (not shown) from the RCMs' aggregated runoff and the reference streamflow (SMP and OBS), shows that CL4 sustains Summer flows, which is a positive aspect. This is not the case for RS4 and PMS, that have steeper recession curves. In addition, RS4 is found to overestimate streamflow peaks in stations from the Duero basin. Finally, we would like to point out that RS4 and PMS depict a different behaviour from the reference data in station 9025 (which defines the Segre sub-basin). Both models overestimate a large number of peaks. We believe that this may be explained by the fact that the Segre sub-basin is nival.



**Table 4.** KGE of the aggregated runoff simulated by the RCMs and LSMs, compared with the reference streamflow of the SIMPA model (upper table) and OBS (lower table). Note that observations are affected by anthropic effects.

| Code | ERA-RS4 | ERA-CL4 | ERA-PMS | ERA-ORC | ERA-ISB | SFR-ISB | Basin | Area (km²) |
|------|---------|---------|---------|---------|---------|---------|-------|-----------|
| 2062 | -0.28 | 0.25 | 0.21 | -0.05 | 0.32 | 0.35 | Duero | 41808 |
| 9011 | 0.54 | 0.05 | 0.28 | -0.17 | -0.12 | 0.57 | Ebro | 40434 |
| 2054 | -0.13 | 0.23 | 0.22 | -0.1 | 0.21 | 0.41 | Duero | 36570 |
| 9002 | 0.6 | -0.01 | 0.33 | -0.2 | -0.19 | 0.56 | Ebro | 25194 |
| 2097 | 0.15 | 0.31 | 0.55 | -0.18 | 0.05 | 0.42 | Duero | 15638 |
| 2043 | 0.23 | 0.29 | 0.57 | -0.19 | 0.02 | 0.31 | Duero | 14283 |
| 9025 | 0.27 | 0.04 | 0.27 | 0.14 | -0.1 | 0.54 | Ebro | 12782 |
| 9120 | 0.52 | -0.03 | 0.19 | -0.28 | -0.19 | 0.58 | Ebro | 12010 |

*(vs. SMP)*

| Code | ERA-RS4 | ERA-CL4 | ERA-PMS | ERA-ORC | ERA-ISB | SFR-ISB | SMP |
|------|---------|---------|---------|---------|---------|---------|-----|
| 2062 | -0.57 | 0.4 | 0.33 | 0 | 0.51 | 0.3 | 0.52 |
| 9011 | 0.61 | 0.14 | 0.43 | -0.15 | -0.07 | 0.79 | 0.52 |
| 2054 | -0.44 | 0.37 | 0.35 | -0.07 | 0.36 | 0.36 | 0.56 |
| 9002 | 0.7 | 0.1 | 0.51 | -0.17 | -0.13 | 0.79 | 0.50 |
| 2097 | 0.13 | 0.36 | 0.59 | -0.19 | 0.08 | 0.45 | 0.82 |
| 2043 | 0.23 | 0.34 | 0.61 | -0.19 | 0.05 | 0.36 | 0.78 |
| 9025 | -0.13 | 0.19 | -0.04 | 0.28 | -0.02 | 0.46 | 0.51 |
| 9120 | 0.53 | 0 | 0.22 | -0.33 | -0.19 | 0.64 | 0.76 |

*(vs. OBS)*

### 5.3.2 SSI and SRI comparison

In this section we study the RCMs' capacity to simulate hydrological drought. Table 5 shows the RMSD and the Pearson correlation coefficient, $r$, comparing RCMs' and LSMs' SRI with SMP's SSI. A colour scale is included to facilitate its reading (green cells indicate a greater similarity of RCMs and LSMs with SMP than red cells).

5     Results are similar to those from Sect. 5.2.1, since RMSD values are above 0.5 (which often indicates a change in drought category). Therefore, RCM and LSM simulations represent hydrological droughts of a different category from that of the reference dataset. Nevertheless, RCMs show moderate positive correlations with SMP, according to the guideline proposed by Evans (1996). It is interesting to see that, although RCMs are in better agreement with SMP when comparing aggregated runoff with streamflow, they show a poorer behaviour than LSMs when SRI is compared to SSI. This might be due to the way

10  downscaling deals with streamflow variability, which plays an important role in hydrological drought, and according to the results from Table 5, seems to be better represented by LSMs than by RCMs.

   We would like to point out that the worst statistics are shown by station 9025, which is identified in the previous section as the station showing the worst performance. It is highly influenced by the effects of snow, which could explain the high RMSD and weak $r$ values.





**Table 5.** Comparison of the SRI from the RCM and LSM simulations with the SSI from SMP. The upper / lower table shows the RMSD / Pearson correlation ($r$).

| Code | ERA-RS4 | ERA-CL4 | ERA-PMS | ERA-ORC | ERA-ISB | SFR-ISB |
|------|---------|---------|---------|---------|---------|---------|
| 2062 | 1 | 0.82 | 0.98 | 0.83 | 0.69 | 0.75 |
| 9011 | 0.78 | 0.87 | 0.84 | 0.8 | 0.68 | 0.56 |
| 2054 | 0.99 | 0.84 | 0.97 | 0.82 | 0.69 | 0.72 |
| 9002 | 0.8 | 0.9 | 0.83 | 0.75 | 0.61 | 0.59 |
| 2097 | 0.99 | 0.91 | 0.98 | 0.91 | 0.75 | 0.72 |
| 2043 | 0.99 | 0.92 | 0.99 | 0.92 | 0.76 | 0.8 |
| 9025 | 0.97 | 1.17 | 1.07 | 0.88 | 0.93 | 0.68 |
| 9120 | 1.07 | 0.94 | 0.97 | 0.87 | 0.68 | 0.69 |

*(RMSD)*

| Code | ERA-RS4 | ERA-CL4 | ERA-PMS | ERA-ORC | ERA-ISB | SFR-ISB |
|------|---------|---------|---------|---------|---------|---------|
| 2062 | 0.49 | 0.62 | 0.47 | 0.62 | 0.73 | 0.69 |
| 9011 | 0.68 | 0.57 | 0.6 | 0.64 | 0.74 | 0.83 |
| 2054 | 0.5 | 0.61 | 0.48 | 0.62 | 0.73 | 0.71 |
| 9002 | 0.66 | 0.55 | 0.61 | 0.69 | 0.79 | 0.81 |
| 2097 | 0.5 | 0.54 | 0.47 | 0.54 | 0.68 | 0.71 |
| 2043 | 0.51 | 0.53 | 0.46 | 0.53 | 0.68 | 0.64 |
| 9025 | 0.49 | 0.24 | 0.36 | 0.57 | 0.52 | 0.74 |
| 9120 | 0.43 | 0.5 | 0.47 | 0.58 | 0.74 | 0.73 |

*(r)*

### 5.3.3 Propagation to hydrological drought

Table 6 shows the $n_x$ values that indicate the month scale at which the correlation between SPI-$n_x$ and i) SRI-1 (RCMs and LSMs), ii) SSI-1 (SMP and OBS) is maximum. This can be interpreted as the temporal scale at which meteorological drought propagates to hydrological drought. To better understand these results and to what extent they reflect the propagation of a

precipitation anomaly to a streamflow anomaly, a colour scale is included to indicate the strength of the correlation. It follows the guideline given by Evans (1996): very strong (blue), strong (green), moderate (white), and weak (red).

LSMs show low $n_x$ values, being the mean $n_x$ 1 (ERA-ORC and ERA-ISB) and 2 (SFR-ISB). Those for station 9025 stand out, since they are 3 and 4 for ERA-ORC and SFR-ISB, respectively. In the first case, it can be explained by the slower dynamics of ORCHIDEE with respect to ISBA, while in the second case, it may be due to the forcing used. SFR is observation

based, build with a dense pluviometric network, and thus, closer to reality than ERA.

RCMs show slower dynamics than LSMs, possibly due to downscaling. Opposite to Sect. 5.2.2, RS4 and PMS show very similar scales: means of 3 months for stations in the Duero basin and 2 months (RS4) and 1 month (PMS) for stations in the Ebro basin. The CL4 provides larger scales: from 9 to 13 months (Duero basin) and from 1 to 8 months (Ebro basin). The difference in scales shown by RCMs is an indicator of the relevance of model formulation in drought propagation. This result

is also obtained in Sect. 5.2.2.





**Table 6.** $n_x$ scale maximizing the correlation between SPI-$n_x$ and i) SRI-1 (RCMs and LSMs), ii) SSI-1 (SMP and OBS). Scales longer than 12 months are marked in bold. A colour scale has been included to indicate the strength of the correlation between SPI-$n_x$ and SRI-1 / SSI-1, following the guide proposed by Evans (1996): very strong (blue), strong (green), moderate (white), and weak (red).

| Code | ERA-RS4 | ERA-CL4 | ERA-PMS | ERA-ORC | ERA-ISB | SFR-ISB | SMP-SMP |
|------|---------|---------|---------|---------|---------|---------|---------|
| 2062 | 3 | 12 | 3 | 1 | 1 | 2 | 9 |
| 9011 | 2 | 8 | 1 | 1 | 1 | 2 | 2 |
| 2054 | 3 | 9 | 3 | 1 | 1 | 2 | 9 |
| 9002 | 2 | 8 | 1 | 1 | 1 | 2 | 1 |
| 2097 | 3 | 13 | 3 | 1 | 1 | 2 | 8 |
| 2043 | 3 | 13 | 3 | 1 | 1 | 2 | 8 |
| 9025 | 2 | 1 | 1 | 3 | 1 | 4 | 2 |
| 9120 | 3 | 6 | 3 | 1 | 1 | 2 | 1 |

| Code | ERA-OBS | SFR-OBS | SMP-OBS |
|------|---------|---------|---------|
| 2062 | 10 | 9 | 9 |
| 9011 | 2 | 3 | 3 |
| 2054 | 10 | 9 | 9 |
| 9002 | 2 | 2 | 3 |
| 2097 | 5 | 10 | 10 |
| 2043 | 11 | 10 | 10 |
| 9025 | 8 | 7 | 8 |
| 9120 | 3 | 3 | 3 |

SMP shows values that range between 8 and 9 months in the Duero basin, and between 1 and 2 months in the Ebro basin.

It can be seen that RCMs and SMP show higher $n_x$ values, and thus slower dynamics, in the Duero basin than in the Ebro basin. This is in agreement with the $n_x$ values obtained using the SSI-1 computed with OBS, where the mean $n_x$ are 9 (Duero) and 4 (Ebro).

Analysing these results we can establish that the RS4 and PMS runoff respond fast to a precipitation anomaly. When compared to SMP and OBS, the month scales provided are in good agreement in the Ebro basin, but too low in the Duero basin. On the contrary, CL4 shows larger scales and behaves inversely to RS4 and PMS, since it is in better agreement with SMP and OBS in the Duero basin.

    ISB shows very strong correlations with SPI-$n$, which confirms the high dependance of streamflow variability with precipi-
10 tation and explains the fast response of ISB to it. ORC and SMP show strong positive correlations, indicating that precipitation has a significant role in streamflow variability. In fact, Quintana-Seguí and Barella-Ortiz (in review) show that SIMPA is more correlated with precipitation than an observational based reference dataset. Regarding the RCMs, RS4 (coupled to ISB) also shows strong correlations in six of the eight stations analyzed, and thus, precipitation is a relevant variable in streamflow variability too. However, correlations of CL4 and PMS with SPI-$n_x$ are moderate (even weak for the 9025 station), which implies
that there are also other factors driving this variability.





## 6 Discussion

This study provides an assessment of how RCMs represent meteorological, soil moisture, and hydrological drought, as well as the way in which a precipitation anomaly propagates to a soil moisture and streamflow anomaly. It is performed using standardized indices, which we believe are a good option to perform drought analysis. The reason being that they describe drought based on the variability of a given variable, and thus, each type of drought can be studied using the variable that best suits its characteristics. In this work, four indices are computed, namely SPI (precipitation), SSMI (soil moisture), SSI (streamflow), and SRI (total runoff).

RCMs are seen to provide a good representation of meteorological drought. Results show that they are capable of reproducing the same drought spells as those detected by their driving data and the reference data. However, they differ between them in the event's duration, intensity, and extension, as expected. We have identified that RCMs improve drought representation with respect to the driving data in several aspects. For instance, the temporal evolution of SPI-12 shows that some of the drought spells' severity is closer to the reference data. In addition, they do not reproduce the spurious trend identified in the driving data which could lead to a misrepresentation of the phenomenon. Finally, a temporal correlation analysis shows that drought representation is ameliorated over the northeastern region of the Iberian Peninsula, which is a known limitation of global analysis over Spain. These results are consistent with previous studies showing that RCMs provided a suitable representation of drought using drought indices over Spain (Barrera-Escoda et al., 2013; Maule et al., 2012; García-Valdecasas Ojeda et al., 2017).

Unlike the previous analysis, results regarding soil moisture and hydrological drought representation show large differences when RCMs are compared between them and to the reference data. The analyses are carried out using the RMSD and the Pearson correlation and the high uncertainty observed corresponds, in most cases, to a change in the drought category of RCMs with respect to the reference data. These differences are expected if we consider the following aspects:

In the first place, the reference data employed. LSM offline simulations are used as reference datasets for the soil moisture and hydrological drought analyses due to the lack of observations. In Spain, soil moisture data from the REMEDHUS network (Martínez-Fernández et al., 2013) and the Valencia Anchor Station (Coll-Pajarón, 2017) are available. However, these sets are suitable for studies where a large spatial coverage is not an issue (like model or satellite-derived data validation), which is not our case. Remote sensing products could be an option too, but there are certain limitations that should be taken into account. For example, uncertainty sources, gaps in the data, and short time series (AghaKouchak et al., 2015). Nevertheless, improvements in this discipline, as well as the increase in the length of time series, will convert these products in an interesting alternative to LSM simulations. Apart from LSM offline simulated total runoff, observed streamflow in mainland Spain is available, and therefore, included in the hydrological drought study. However, the large number of dams and canals have a high anthropic impact on its river systems, affecting observed streamflow. Since RCMs do not take into account these effects, a simulation from the SIMPA model, that provides streamflow in a natural regime is neccesary as a reference dataset. It would be interesting to include these effects in RCM modelling to perform the drought analysis using observed streamflow as reference data. This would provide an idea of the anthropic impact on hydrological drought.



In the second place, the use of LSMs. One of their main characteristics is that they simulate the water cycle by means of physical principles, which converts them in an interesting tool to study both soil moisture and hydrological droughts. However, an important drawback is that soil moisture is modelled differently in each LSM (Koster et al., 2009). This affects our analysis in two ways. First, the reference datasets will vary depending on the LSM chosen and its modelling approach. That is the

reason for using three soil moisture reference data sets in this study. Second, the RCM's soil moisture will also depend on the LSM modelling, as it is coupled to the LSM. Streamflow modelling is another drawback in LSMs, since it is modelled at coarse resolutions. In addition, there are other issues to consider. For example, LSMs may lack of processes related to streamflow, like hortonian runoff, underground water and horizontal water transfer. Actually, the ISBA-3L LSM, does not take them into account.

In the third place, the use of simulated total runoff to compute SRI due to the lack of RCMs' modelled streamflow to compute SSI. Hydrological drought representation and propagation are likely to be affected by this approximation according to the KGE values shown in Sect. 5.3.1. When compared to SMP, the KGE using total runoff is under 0.25 in almost half of the cases, which indicates a poor performance. However, we would like to stress that PMS is the RCM that approximates better streamflow according to the KGE values.

Finally, the effects of RCM coupling with the atmosphere should be taken into account, since the soil moisture and hydrological references are LSM simulations performed in an uncoupled mode, and thus do not have the atmosphere's feedback.

We would also like to point out the relevance of the time scale used to compute drought indices. According to García-Valdecasas Ojeda et al. (2017) and Bowden et al. (2016), the added value of the SPI is influenced by the time accumulation period employed, and RCMs would increase it at longer time scales. This is interesting to consider because soil moisture and

hydrological droughts have different scales of propagation, but also because of the large differences found in the propagation analysis from Sect. 5.2.2 and 5.3.3.

This study analyses three RCMs, out of which the RS4 shows larger similarities with the reference data and deviates less from its driving data in the meteorological and soil moisture drought analyses, while PMS shows better performance regarding the hydrological drought. Nevertheless, it should be stressed that differences in soil moisture and hydrological drought charac-

terization are still relevant as the three RCMs represent drought categories which differ from those provided by the reference data.

A key result of the study is the relevance of the models' physics, which prevails over the driving data. This is showed in the soil moisture and hydrological drought representation studies, as well as in the analyses of drought propagation. In the latter case, the model's formulation influences the temporal scale at which the variability of precipitation affects that of soil moisture

and streamflow. For instance, in the analysis from Sect. 5.2.2, the spatial patterns of RS4 (coupled to ISB) and ISB, are very similar between them and show that ISB, and thus RS4, respond too quickly to precipitation. The spatial structures of PMS (coupled to ORC) and ORC differ more between them. PMS' temporal scale is shorter than that of ORC, which may be due to coupling effects and to the fact that PMS' precipitation extremes are too strong (Domínguez et al., 2013).




The results obtained are coherent with those from Quintana-Seguí and Barella-Ortiz (in review), where the soil moisture and hydrological drought analyses showed large differences between LSMs too. Therefore, both studies give clear proof that improvements in modelling concerning soil moisture and streamflow are needed.

# 7 Conclusions

In a context of a changing climate it is necessary to know the evolution of extremes, like drought. Understanding the processes involved is, therefore, vital. For this, the current modelling tools must be first evaluated. The work here presented analyses how RCMs represent meteorological, soil moisture and hydrological drought, together with the propagation from a precipitation anomaly to soil moisture and streamflow anomalies.

It is concluded that RCMs provide added value to meteorological drought representation, avoiding possible error sources
from the driving data and ameliorating its characterization over areas which are known to pose certain problems to global driving data products. However, soil moisture and hydrological drought representation by RCMs show large uncertainties. This is manly due to the relevance of model physics and its prevalence to the driving data. Similar results are obtained for the propagation processes, where model formulation influences the dynamics of drought propagation, showing different temporal scales depending on how precipitation variability is formulated in the model.

In our opinion, RCMs are a suitable tool for meteorological drought studies, but should be use cautiously for soil moisture and hydrological drought analyses. Improvements regarding soil moisture modelling and streamflow related processes (natural and anthropic) should be performed for a better characterization of drought events, as well as their propagation.

Some perspectives to this work are to extend the analysis by including more RCMs and other drought indices. On the one hand, using more models would increase the information about drought simulation at a regional scale, but also allow to identify
further improvements in LSMs. On the other hand, if the analysis is performed with other indices, we can study the effects that other variables and processes, like temperature and evapotranspiration have on droughts. Studies of seasonal effects on droughts would also be interesting as these effects play an important role in drought propagation. Finally, it would be interesting to analyse the drought indices added value as a function of the time scale used and how this may affect drought representation and its propagation.

*Data availability.* The forcing datasets and driving data used in this study can be accessed from their original source: ERA-Interim: https://www.ecmwf.int/en/forecasts/datasets/archive-datasets/reanalysis-datasets/era-interim; the SAFRAN dataset for Spain is available for research purposes from the Mistrals-HyMeX database (Quintana-Seguí, 2015). The RCM simulations were downloaded from the Med-CORDEX database: https://www.medcordex.eu/. The ORCHIDEE and SURFEX LSM simulations were produced for this study, but can be reproduced using the corresponding release of the models: https://forge.ipsl.jussieu.fr/orchidee (release no. 4676); SURFEX: https://www.umr-
cnrm.fr/surfex/spip.php?rubrique. Finally, observed and modelled streamflow from SIMPA can be accessed from the Spanish Ministry for the Ecological Transition website: https://www.miteco.gob.es/en/cartografia-y-sig/ide/descargas/agua/default.aspx





*Competing interests.* The authors declare that they have no conflict of interest.

*Disclaimer.* TEXT

*Acknowledgements.* We thank the Spanish State Meteorological Agency (Agencia Estatal de Meteorologia, AEMET) and the Spanish Ministry for the Ecological Transition for providing us with data. This work is a contribution to the HyMeX program (Hydrological Cycle in the Mediterranean Experiment; http://www.hymex.org). It has been funded by the Spanish Ministry of Science, Innovation and Universities and the European Regional Development Fund through the MARCO (CGL2013-47261-R) and HUMID (CGL2017-85687-R) projects, and by the Interreg POCTEFA project PIRAGUA (EFA210/16/ PIRAGUA).



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
