# Peer review of "Evaluation of drought representation and propagation in Regional Climate Model simulations across Spain"

_Hydrology and Earth System Sciences, 2018_

## Referee Comment (RC1) · Anonymous Referee #1 · 11 Jan 2019

Title: Evaluation of drought representation and propagation in Regional Climate Model simulations over Spain.

Authors: Anaïs Barella-Ortiz and Pere Quintana-Seguí.

Recommendation: reject/resubmit

Summary:

This paper evaluates the meteorological and hydrological drought (here including soil moisture) representation using the standardized indices (SPI, SSMI, SRI, and SSI) in Spain. Moreover, meteorological drought propagations into hydrological droughts (soil

moisture, runoff, and streamflow) were analyzed by correlating SPI-x (x=accumulation periods) with hydrological drought indices (SSMI-1, SRI-1, and SSI-1). 2 forcing data (SAFRAN and ERA-i), 6 models (RS4, CL4, PMS, ISB, ORC, and SMP), and discharge observations were used in the study. They conclude that RCMs is better suited for meteorological drought analysis than hydrological ones. In addition they also suggest to use more models and indices.

Assessment:

The topic of the paper is fit well in HESS and is of great interest for the HESS readers. The findings are also interesting for readers working on the drought and modeling of climate and land surface models. While the work and findings are interesting, the paper is poorly written especially in the abstract, introduction, data, and method sections. I have a problem in understanding some sentences and statements. The conclusion does not describe the important findings of this paper. Based on my review, I decide to reject the paper at this state. However, I encourage authors to reconsider to resubmit their work. I provide my comments below and would ask the authors to take these comments into consideration as they resubmit the paper.

General comment:

1. The title is misleading. This study is not only used RCMs but also LSMs and hydrological models (HMs).

2. The abstract is not informative and poorly written. The objective of this study is not clear. For example authors stated that it is vital to study the evolution of drought in relation to climate change, and therefore better understanding processes involved in "it" is a key. This is not your objective to study the evolution of drought related to climate change. Second, the methods and tools that authors used in the study are not well defined in the abstract. Last, the conclusion does not summarize the main findings.

3. The way the authors wrote their introduction can be concluded as follows: 1) many sentences in the introduction part are unclear and need to be re-written, 2) there is no clear story line, 3) the introduction is not well structured in reasoning, 4) missing many related references e.g., increase of drought in Mediterranean due to climate change, drought propagation, models uncertainty from WATCH. 4. Section 2 and section 3 can be combined and need to be restructured. For example: 1) in section 2 the authors already discuss about the precipitation amount simulated by the models. 2) Section 3.2 discusses about the models, but LSM, RCM, and HM models are also models. So why authors separated them? 3) SURFEX and ORCHIDEE are LSM and why authors wrote them in different sub sections than LSM section (3.2.1). 4) I suggest the authors start with RCMs first and then followed by LSMs and HMs.

5. Section 5 and 6 are well structured and also better in writing than other aforementioned sections. The conclusion is also expected. Of course the meteorological drought is better represented using climate model, and hydrological drought better represented using hydrological models. Moreover, authors also highlight the use of more models and indices, where in my opinion it is not necessary. This paper already used many models (RCMs and HMs). Different indices can be used but it depends on the goal of the study.

6. Some typos and grammar mistakes found in the texts.

Line by line comment: L refers to Line and P refers to Page Note: Authors still need to pay attention for other sentences that are not mentioned here.

L171: Already in line 1 I do not understand with your sentence. Here you stated: "Drought is an important climatic risk, of modeling.." what do you mean with that? Rephrase the sentence.

L4P1: You may remove "comma" from: "The study here presented, analyses.."

L5P1: here you only mention RCM models. How about LSMs and HMs that you used?

L6-7P1: You stated: "...these models improve meteorological drought representation, but..." It is unclear how you tested this? It was not stated in the previous sentences.

L8P1: You said: "These are mainly due to the relevance of model formulation" Is it model formulation or structure?

L13-14P1: You stated: "...impacts from recent climate related extremes show a significant vulnerability..." I do not understand this. Impacts from climate extremes show vulnerability?

L15P1: Do you mean different development level such as less develop and well develop countries?

L18P1: Limited references. There are many.

L22P1: There are more references for drought types, e.g., Van Loon (2015): hydrological drought explained, Mishra and Singh (2010): A review of drought concepts.

L4P2: What do you mean with there are no thresholds for soil moisture drought?

L6P2: Runoff is also one of hydrological drought where you also used in this study.

L9P2: Here you introduce environmental drought. Can you give references about that? As far as I know, it is socio-economic drought.

L10P2: You may add Van Loon et al. (2012): Evaluation of drought propagation in an ensemble mean of large-scale hydrological models.

L12P2: I do believe it is not potential evapotranspiration but actual evapotranspiration.

L12-13P2: This statement: "....related to the availability of irrigation water in irrigated areas, thus depending on the hydrological drought" is unclear.

L15P2: You may write "some years"

L15-16P2: You wrote: "However, if there is no further delay in precipitation, it can occur that no hydrological drought is observed" What do you mean? Small precipitation

amount may not alleviate hydrological drought.

L19-20P2: Repetition with first paragraph.

L21-22P2: It is not necessary if you analyze drought without human influence where this may be the case in the models that you used.

L28P2: You may remove the word "used"

L32-33P2: It is not clear sentence: "following this line". Also you may replace ";" with ","

L4-5P3: I do not agree that RCM simulation is cheaper than GCM in terms of computing resources. First RCM needs GCM outputs for boundary model's inputs, second the computing resources depends on the resolutions of the model.

L13-14P3: Unclear sentence. What kind of tools and what will be improved?

L15P3: You may remove ","

L15P3: you may replace "capacity" with "capability"

L17P3: Too many "it" can you define what are it?

L19P3: Should be analyzed since they did the studies in the past.

L20P3: Replace "uses" with "used" and also define those three drought indices. What are they?

L21P3: Replace use(s) with used. The word "too" is informal.

L23P3: Replace analyse with analysed.

L25P3: Replace use and study with their past tense form.

L26P3: you may replace: "..studies are…" with "…those studies were…"

L27P3: Replace analyse with analysed.

L28P3: Replace are with were.

L29-32P3: Why don't you combine these two sentences?

L6P4: Missing "the". The standardized indices.

L7P4: You may replace "with" with "using"

L8-10P4: Please write the full meaning of SSMI, SSI, and SRI first since this is the first time you introduce these acronyms.

L11P4: Rewrite: "...we will detail the study are".

L18-25P4: This paragraph should be in the result section.

L26-27P4: Can you elaborate more why soil moisture anomaly in spring may influence droughts and heatwaves? Also this statement is not belonging to study area.

L29P3: I suggest you to use different word for relief such as mountainous areas?

L31P4: What is "it" refers to?

L32P4: Rewrite "to use" into "to be used"

L2P5: You may write: "..a reduction of precipitation..."

L5P5: What is "it" refers to?

L8P5: You write: "...increase this type of drought". In my opinion snow melt can increase or decrease the hydrological drought depend on early or late snow melt.

L17P5: Why don't you combine paragraph 2 with 1?

L23-24P5: This paragraph stands alone and can be combined with previous paragraph.

L28P5: Rewrite: "...detailed in Quintana-Segui et al. (2016).

L31P5: You may replace "in" with "for"

L6P6: Again you may combine all paragraphs in this section (3.1.2) into 1. Paragraph 1 consists of two sentences, which are about ERA-interim and then paragraph two starts with the word "it". You should combine this paragraph since you still discuss about the same thing. Also you cannot start new paragraph with the word "it".

L9P6: What is "it" refers to? Spatial resolution?

L11-12P6: please give reference.

L12-13P6: You write: "…they can be corrected". How?

L1P7: Rewrite this sentence: "Since there is no observed truth for soil moisture available,…"

L8-9P7: You should write the full names first before the acronyms.

L14P7: You may add "in" before 1 km resolution.

L14P8: Rewrite the opening sentence.

L18P8: Replace "is" with "are" since you mention about drought representation and propagation.

L20P8: You may reverse the words into "…represent physical processes in different ways…"

L22P8: You used the name CNRM-RCSM4 but for previous sentence you use the name RCSM4 only. Please be consistent. Also for COSMO-CLM in L27.

L1P9: In the section 3.4, please mention the number stations that you used in your study.

L2-3P9: Missing verb in this sentence.

L3P9: You may write: "To obtain monthly time series…"

L4P9: Can you elaborate why south is not as represented as the north?

L5P9: You may rewrite "to having" with "to have"

L7-8P9: You wrote: "In this way, both studies complement each other. . .." It is not clear sentence. What is in this way?

L10-11P9: This sentence is not clear. You may write: ". . .we compute SPI and SSMI, which require precipitation and soil moisture data, respectively".

L14-16P9: Proper citations for all indices you mention here, except if you already did so before.

L1P10: You said the variable's time series is transformed from its original distribution to a normal one. What kind of distribution? Gamma, GEV, Pearson, etc?.

L3P10: You may rewrite: ". . .in the meteorological case. . ." into "for meteorological case. . .."

L4P10: You may change "for" with "from" and also missing verb.

L8P10: Why only for SPI-12?

L13-14P10: Please give reference.

L17P10: Please use better opening sentence.

L19P10: You may write actual evapotranspiration.

L15P11: You can combine second paragraph with the first.

L17-18P11: What do you mean with relief is a determining factor in distribution?

L21P11: Modeled precipitations usually have higher results than the observed and ERA-i. There are some papers show that.

L4P12: Drought in 2005-2006 does not coincide with those found "in" Belo-Pereira et al. (2011) but it occurred. So please rephrase.

L6P12: It is hardly to see in the Figure. Can you write the number of months?

L2P13: typo for tan

L10P13: You may write: "The difference is the degree to which they deviate"

L11P13: What are "it" refer to?

L12P13: What do you mean with "do it"

L6P15: How about the white color in the table?

L12P15: Please rewrite: "…RCMs compared between them…"

L14P16: Please elaborate more why SPI with higher accumulation period is slower.

L3P17: you may rewrite: "…to the LSM with the surface scheme it is coupled to." Into "…LSM coupled with the surface scheme."

L7P18: Rewrite: "To single out stations"

L19-20P18: I thought LSMs should have better results for hydrological drought than GCMs.

L22P18: I saw 0.6 and not 0.7. Also you said that PMS behaves better over both basins. How do you define PMS is better? By average value or by color?

L4P19: Again what is white?

L14P21: You may change from "the 9025 station" to "station number 9025"

L10P22: What is extension?

L33P23: You may rewrite: "..that PMS' precipitation extremes are too strong" into "…that PMS simulates higher precipitation amount."

L9P24: I believe you cannot avoid the error but you can minimize the error.

L15P24: Replace use with used. Past tense passive sentence.

Figures: They are too small, cannot see them clearly.

[Figure]

Table 6: I cannot find the number in bold. Can you also please write the correlation numbers for each color in the caption?

---

## Referee Comment (RC2) · Anonymous Referee #2 · 30 Apr 2019

**Review: Evaluation of drought representation and propagation in Regional Climate Model simulations over Spain**

Anaïs Barella-Ortiz and Pere Quintana-Seguí

**Summary**

This paper assesses the use of LCMs, RCMs and HMs to investigate drought propagation in Spain. They conclude that one can use RCMs to investigate metrological droughts in Spain, but further work should be undertaken to model soil moisture and hydrological droughts.

This paper is of sufficient novelty and fits well within the scope of HESS. However, in its current form it is difficult to understand what was done, and crucially why it was done and what the applications of the research are. As such, I feel this paper cannot be accepted as is but I provide my comments below which I hope the authors will consider before resubmitting to HESS, I will be happy to re-review this paper upon resubmission.

**Decision: Reconsider on resubmission**

**General comments**

I have not listed detailed line by line comments as I feel more work is needed to address the more general issues given below.

The language throughout makes it difficult to understand at times (for example, P23L18-19: It is not clear if this means that RCMs are not appropriate to calculate SPI at longer accumulation periods) and should be improved dramatically before publication.

The paper lacks clearly defined aims and applications, currently it reads as a modelling exercise rather than science to support real-world applications – this can be addressed by a better structured introduction as it currently jumps around without a coherent story (e.g. what is the problem, what have others done in the past, what is the research gap, what is the aim of this study and how this will address the research gap).The introduction and literature review also relies heavily on the IPCC reference, without reviewing peer reviewed publications (and where papers are introduced, they are often listed as 'other papers on the topic' such as P2 L32) and outlining the research gap this paper is aiming to fill. The lack of aims and disjointed nature of the paper make it difficult to reach the conclusions set out in the final section of the paper.

It has not been made clear why this modelling approach was used. The assertion on P6L20 that the atmospheric feedback not being accounted for makes LSMs  a good tool to study drought because they can be treated like physically distributed hydrological models, does not provide explanation – why do you want them to behave like a physically distributed model?

You don't mention or address the issue of uncertainty – what about the uncertainties of the modelling approach? Could you explore this using a multi-model ensemble?

In many places, there is text seemingly in the wrong section of the paper, for example P3L33-P4L4 should more likely sit in the data/methods section as this is detailed for the introduction. P5L1-8 would be better placed in the introduction. P11L24-27 would be better placed in the discussion describing why there were discrepancies between the modelled outputs. The authors should review the text to ensure that descriptions of data and methods and discussion text are in the appropriate sections.

Section 3.1: this should have more introductory information before diving into the detailed descriptions of SAFRAN and ERA, in 3.1 please outline what variables you use and what they are needed for before describing them in turn.

In Section 3 the RCMs are introduced third but surely start the modelling chain, I suggest you introduce these first, then the LSMs then Hydrological Models. Was it necessary to calibrate and validate your models – how did you do this?

A lot of detail is provided about the LSMs which is published elsewhere and appears to pad the paper, much of the model background can be removed – the focus should be on why the models were chosen and what they will be used for.

Section 3.4: It would be useful to include a map of the observation stations used- how many stations were used? Only 8 across the whole of Spain? Why not more – there must be more than 8 stations that have 95% data completeness?

P18L8: What evidence or scientific literature did you use to select the 'arbitrary' KGE of 0.5? Later in Section 5.3.1 you say performance of CL4 is poor because KGE is generally below 0.5, but the best performance is for RS4 with KGE of 0.7 – is this enough of a difference between poor and best (reading 'good') performance?

Section 5.3.1 – why were the temporal analysis not shown? If you only have 8 gauging stations, it would be simple to include time series plots showing the modelled ensemble data against the observations.

Table 6: This might be better as a figure with the catchment areas coloured by SPI-$n_x$ – as readers we don't know where your catchments are, how do the results vary spatially? What might the effect of catchment properties be on the propagation process? How well do the different models represent these catchment properties?

Table 6: What r values are associated with the Evans classification? How significant are these correlations? The bold type face mentioned in the caption is not obvious in the table.

P22L4: why do you believe standardised indicators are appropriate for this study? This should have been outlined previously.

P22L10: what is meant by event extension? The duration and intensity (and extension) of events is not described elsewhere nor shown in any figures – what do you refer to here?

In general the figures were too small and labels too small to read. You should avoid the red-blue colour schemes of Tables 3-6, they are not appropriate for those who are colour blind. You can check whether your figures are colourblind friendly here: http://www.color-blindness.com/coblis-color-blindness-simulator/ or by using the CVSimulator app

In regards to the Barker et al. (2015), you have cited the Discussions paper, please cite the final 2016 paper (https://www.hydrol-earth-syst-sci.net/20/2483/2016/hess-20-2483-2016.html).

On P3L31 Lopez-Morreno's name has been misspelled.

The tense throughout is the present tense, however, research is conventionally written up in the past tense (as it is work that has been completed), please correct this in the next version of the paper.

---

## Author Response (AR1)

**Point-by-point response to the referees' reviews**

The replies are written in bold type after the replies sent during the open discussion to each referee. The pages and lines numbers indicated in these replies are referred to the marked-up manuscript version.

**Referee 1**

Dear Referee,

Thank you very much for taking the time to read the manuscript and commenting on parts of it. We are pleased that you consider both the topic and findings interesting for HESS readers. As for the way it is written, we agree with you that the presentation of this research can be clarified. Below you will find the answers to the comments you made about the manuscript titled "Evaluation of drought representation and propagation in Regional Climate Model simulations over Spain" by A. Barella-Ortiz and P. Quintana-Seguí.

Once the other referee's comments are available, we will study them and see to complement the modifications proposed by this referee with yours.

**General comments:**

1. The title is misleading. This study is not only used RCMs but also LSMs and hydrological models (HMs).
   You are right in that the study uses LSMs and a hydrological model besides RCMs, but simulations from these are used as references to evaluate how RCMs represent droughts and the way precipitation anomalies propagate to soil moisture and streamflow anomalies. This is explained in the Introduction section (page 4, lines 1-5) and in Sect. 3 (page 7, lines 19 and 32 and page 8, line 11). Nevertheless, we will make it clearer in Sect. 3 (Datasets). In our opinion it is not necessary to modify the title as the evaluation of drought representation and propagation is performed for RCMs, but if the referees and the editor deem it necessary, we can think about an alternative.
   **Modification:**
   **Following the advice of the Scientific Editing Service, the title has been changed to "Evaluation of drought representation and propagation in regional climate model simulations across Spain"**

2. The abstract is not informative and poorly written. The objective of this study is not clear. For example authors stated that it is vital to study the evolution of drought in relation to climate change, and therefore better understanding processes involved in "it" is a key. This is not your objective to study the evolution of drought related to climate change.

Second, the methods and tools that authors used in the study are not well defined in the abstract. Last, the conclusion does not summarize the main findings.
We will rewrite the abstract indicating more clearly the main problematic and reason why we performed the study, the methodology employed, the RCMs analysed as well as the LSMs and hydrological model used as reference data, and the conclusions.
**Modification:**
**The abstract has been rewritten.**

3. The way the authors wrote their introduction can be concluded as follows: 1) many sentences in the introduction part are unclear and need to be re-written, 2) there is no clear story line, 3) the introduction is not well structured in reasoning, 4) missing many related references e.g., increase of drought in Mediterranean due to climate change, drought propagation, models uncertainty from WATCH.
We agree with you that the introduction should be clarified. We will restructure it in order to make clear the story line and reasoning. More references will be added too.
**Modification:**
**Sec. 1 (Introduction) has been rewritten and clarified.**

4. Section 2 and section 3 can be combined and need to be restructured. For example: 1) in section 2 the authors already discuss about the precipitation amount simulated by the models. 2) Section 3.2 discusses about the models, but LSM, RCM, and HM models are also models. So why authors separated them? 3) SURFEX and ORCHIDEE are LSM and why authors wrote them in different sub sections than LSM section (3.2.1). 4) I suggest the authors start with RCMs first and then followed by LSMs and HMs.
We wrote separately sections 2 and 3 because  the first one describes the study area and the second one, explains i) the forcing and driving datasets and ii)  the models (RCM, LSM, and HM) used in this study. We think that this separation is necessary and reasonable, but if the editor thinks that they must be fusioned, we can find a way to put them together.
   1. The reason why precipitation amount is discussed in Sect. 2 is to provide the reader with the general behaviour of the precipitation regime in Spain. This is key to analyse meteorological drought and its propagation. However, we agree with you that there might be redundant information regarding the precipitation regime in Sects. 2 and 5.1.1. We will rewrite them to avoid this issue.
   2. We agree with the author that Sects. 3.2 and 3.3 should be unified. There is a numbering error.
   3. This is similar to the previous point, there is a numbering error and Sects. 3.2.2 and 3.2.3 should be sub-sections of Sect. 3.2.1.
   4. We will follow your advice and restructure Sect. 3  as follows:
       3 Datasets
           3.1 Forcings and driving data
               3.1.1 SAFRAN meteorological analysis
               3.1.2 ERA-Interim

3.2 Models
    3.2.1 Regional climate models
    3.2.2 Land Surface Models
        SURFEX
        ORCHIDEE
    3.2.3 Hydrological models
3.3 Observations

SURFEX and ORCHIDEE sub-sections will not be numbered, because only three levels of sectioning are allowed.

**Modification**

**The structure of the manuscript has been modified. However, we did not follow the structure proposed previously in this comment. Section 3 (Datasets) (P5) has been modified in the following way:**

**3 Datasets**
    **3.1 Regional climate models**
    **3.2 Reference products**
        **3.2.1 Meteorological drought reference**
        **3.2.2 Soil moisture drought reference**
        **3.2.3 Hydrological drought reference**

**We believe that this structure clarifies that RCMs have been analysed and SAFRAN, offline LSM simulations, SIMPA, and observed streamflow have been used as reference data. In addition, Tables 1 and 2 have been modified. Table 1 provides the RCMs analysed in the study and Table 2 lists the reference data used in the study.**

5. Section 5 and 6 are well structured and also better in writing than other aforementioned sections. The conclusion is also expected. Of course the meteorological drought is better represented using climate model, and hydrological drought better represented using hydrological models. Moreover, authors also highlight the use of more models and indices, where in my opinion it is not necessary. This paper already used many models (RCMs and HMs). Different indices can be used but it depends on the goal of the study. We are pleased that you find these sections better structured.

We think that your phrase "Of course the meteorological drought is better represented using climate model, and hydrological drought better represented using hydrological models" does not correctly describe the results of our study. Three types of drought (meteorological, soil moisture, and hydrological) are analysed, not two types. In addition, besides drought representation, we also analyse drought propagation: from a precipitation anomaly to soil moisture and streamflow anomalies. Finally, it is important to bear in mind that the study is focused on drought representation and propagation by RCMs and that the other types of models (LSMs and hydrological) are used as reference.

Regarding the perspectives, we believe that it is interesting to analyse more RCMs, because it helps to better understand uncertainties and improve modelling. As for the indices, we provide two examples that we believe are of interest to improve drought knowledge and modelling, one referred to a variable (temperature) and another one referred to a process (evapotranspiration). Both have an important impact on drought, but it is true that this also depends on the objectives. In our case, the objectives are to understand drought in a way as complete as possible.

**Modification:**

**We hope that after the changes performed in the manuscript, the objectives, datasets, and methodology are clearer.**

6. Some typos and grammar mistakes found in the texts.
   Line by line comment: L refers to Line and P refers to Page Note: Authors still need to pay attention for other sentences that are not mentioned here.
   We would like to thank you for pointing out the typos and grammar mistakes you found in the text. We will send the manuscript to a Scientific Editing Service to assure that the manuscript's English is correct.

   **Modification:**

   **The manuscript has been sent to a Scientific Editing Service.**

Below you will find the replies to the line by line comments.

1. L171: Already in line 1 I do not understand with your sentence. Here you stated: "Drought is an important climatic risk, of modeling.." what do you mean with that? Rephrase the sentence.
   The objective of the phrase is to inform that drought is an important climatic risk (which is expressed in a large number of scientific papers) and that drought is difficult to model because the interaction of several processes (atmospheric and continental), with different temporal scales, has to be taken into account. We will rephrase the sentence.

   **Modification:**

   **The abstract's first paragraph has been rewritten as "Drought is an important climatic risk that is expected to increase in frequency, duration, and intensity as a result of a warmer climate. It is complex to model due to the interactions between atmospheric and continental processes. A better understanding of these processes and how the current modelling tools represent them and characterize drought is vital" (L1-3P1).**

2. L4P1: You may remove "comma" from: "The study here presented, analyses.."
   The comma will be removed.

   **Modification:**

   **The abstract has been rewritten and the beginning of the phrase has been modified to "The aim of this study is…" (L4P1)**

3. L5P1: here you only mention RCM models. How about LSMs and HMs that you used?
RCMs are mentioned because they are the models that we have evaluated in this work. However, as explained in the "General comments", number 2 reply, the abstract will be rewritten and the models whose simulations are used as references, will be mentioned.
**Modification:**
**The datasets used as reference (SAFRAN, LSM offline simulations from ISBA-3L and ORCHIDEE, and a SIMPA hydrometeorological model simulation and observed streamflow) are included in the abstract (L10-12P1).**

4. L6-7P1: You stated: ". . .these models improve meteorological drought representation, but. . ." It is unclear how you tested this? It was not stated in the previous sentences.
This was tested by means of standardized indices computed using variables directly related to each type of drought:
   ● SPI index computed using precipitation (meteorological drought).
   ● SSMI index computed using soil moisture (soil moisture drought).
   ● SRI and SSI index computed using runoff and streamflow, respectively (hydrological drought).
The comparison between the indices provided by i) RCMs and ii) the reference data was performed by means of the RMSD and the Pearson correlation.
As explained in the "General comments", number 2 reply, the abstract will be rewritten and the methodology will be informed.
**Modification:**
**The text from the abstract informing about the methodology has been extended: "The analysis was carried out by means of standardized indices calculated using variables directly related to each type of drought: precipitation (SPI), soil moisture (SSMI), runoff (SRI), and streamflow (SSI)" (L5-7P1).**

5. L8P1: You said: "These are mainly due to the relevance of model formulation" Is it model formulation or structure?
We mean model structure. We would like to thank you for pointing this out.
**Modification:**
**"Formulation" has been replaced by "Structure" (L15P1).**

6. L13-14P1: You stated: ". . .impacts from recent climate related extremes show a significant vulnerability. . ." I do not understand this. Impacts from climate extremes show vulnerability?
No, the ecosystems and human systems are vulnerable, as the phrase mentions. Maybe the phrase could be rewritten for better clarity. In fact, this is mentioned in the Intergovernmental Panel on Climate Change (IPCC) Fifth Assessment Report.
**Modification:**
**The phrase has been rewritten as "Impacts from recent climate-related extremes show that ecosystems and human systems are significantly vulnerable and exposed to current climate variability (IPCC, 2014)" (L20-21P1).**

7. L15P1: Do you mean different development level such as less develop and well develop countries?

Yes, we refer to well developed, developing, and underdeveloped countries. We will rephrase the sentence to make it clear.

**Modification:**

**This text has been removed from the Introduction section to shorten and clarify it.**

8. L18P1: Limited references. There are many.

More references will be added.

**Modification:**

**The reference Hoerling et al. (2012) has been included.**

9. L22P1: There are more references for drought types, e.g., Van Loon (2015): hydrological drought explained, Mishra and Singh (2010): A review of drought concepts.

We would like to thank you for suggesting these references. In fact they are already given in the text. Following your advice, we will also include them in this sentence.

**Modification:**

**We have included Mishra and Singh (2010) and Van Loon (2015) in this sentence (L3P2).**

10. L4P2: What do you mean with there are no thresholds for soil moisture drought?

There are no standard determined values that once they are exceeded imply that there is a soil moisture drought. This is because this type of drought depends on other aspects, like the soil's type of vegetation or crop, its location, and season. But we agree that it is not clear now and we will remove this phrase from the text.

**Modification:**

**The phrase has been removed from the text.**

11. L6P2: Runoff is also one of hydrological drought where you also used in this study.

We would like to thank you for pointing this out. Runoff will be included in the hydrological drought description.

**Modification:**

**Runoff has been included in the hydrological drought description (L8P2).**

12. L9P2: Here you introduce environmental drought. Can you give references about that? As far as I know, it is socio-economic drought.

The Intergovernmental Panel on Climate Change (IPCC) Fourth Assessment Report. WGII: Impacts, Adaptation, and Vulnerability lists the four types of drought given in the manuscript. Regarding environmental drought, it says that it is a combination of the other three types. We would like to thank you for pointing this out, since the reference provided is not the correct one. It should be IPCC (2007).

**Modification:**

**The reference IPCC (2014) has been corrected to IPCC (2007) (L3P2). The environmental drought has been removed from the text to make it more concise.**

13. L10P2: You may add Van Loon et al. (2012): Evaluation of drought propagation in an ensemble mean of large-scale hydrological models.

We would like to thank you for suggesting this references. In fact it is already given in the text. Following your advice, we will also include it in this sentence.
**Modification:**
**The reference Van Loon et al. (2012) has been included (L11P2).**

14. L12P2: I do believe it is not potential evapotranspiration but actual evapotranspiration.

We will replace "potential" by "actual".
**Modification:**
**Potential" has been replaced by "actual" (L13P2).**

15. L12-13P2: This statement: ". . ..related to the availability of irrigation water in irrigated areas, thus depending on the hydrological drought" is unclear.

Our purpose is to show that the different types of drought are related between them in different ways. A key factor defining the relation of soil moisture drought with the other types is irrigation.

- Non irrigated areas: soil moisture drought appears slightly out of phase of the meteorological drought and depends on several factors (soil type, capacity to retain water, and evapotranspiration).
- Irrigated areas: we can distinguish two cases:
  - In purely irrigated areas, soil moisture drought is directly related to the availability of irrigation water and, therefore, it depends on hydrological drought.
  - In regions where irrigation is carried out under unusual circumstances, to avoid crop loss, soil moisture drought is not due to a lack of irrigation water, but of the water content in the soil. In this case, soil moisture drought is related to meteorological drought.

We will rewrite the text to make it more clear.
**Modification:**
**The text has been extended: "This is also applicable to regions where irrigation is carried out under unusual circumstances, for example, to avoid crop loss. In irrigated areas, soil moisture drought is directly related to the availability of irrigation water and therefore depends on hydrological drought" (L13-15P2)**

16. L15P2: You may write "some years"

Following your advice, we will replace "year" by "years".
**Modification:**
**The text has been removed from Sec. 1 to shorten and clarify the section.**

17. L15-16P2: You wrote: "However, if there is no further delay in precipitation, it can occur that no hydrological drought is observed" What do you mean? Small precipitation amount may not alleviate hydrological drought.

After re-reading the paragraph we think that it should be simplified, as it is not clear enough. What we mean is that soil moisture and hydrological drought are related to meteorological drought, but that meteorological drought alone does not explain the other kinds of drought.

**Modification:**

**The text has been removed to avoid confusion.**

18. L19-20P2: Repetition with first paragraph.

The sentence will be removed from this paragraph.

**Modification:**

**The paragraph has been removed for the sake of clarity.**

19. L21-22P2: It is not necessary if you analyze drought without human influence where this may be the case in the models that you used.

We agree with you that if the models used to analyse drought representation and propagation do not consider anthropization, human action does not have to be taken into account in the drought modelling process. However, we believe that it is important to include this information. The reason being that, at this stage of the Introduction section, we are talking about modelling drought from a general perspective, without being restricted to the RCMs' characteristics. But this passage can be removed if necessary.

**Modification:**

**Although the phrase has not been removed, it can be deleted if the Editor and referees deem it necessary.**

20. L28P2: You may remove the word "used"

The word "used" will be removed.

**Modification:**

**The text has been rewritten.**

21. L32-33P2: It is not clear sentence: "following this line". Also you may replace ";" with ","

We will modify the sentence to "Other studies following this line of improvement are:...".
We will also replace the semicolon by a comma.

**Modification:**

- **The sentence has been modified: "Other studies analysing the effect of the accumulated period are Vicente-Serrano and López-Moreno (2005) and Edossa et al. (2009)" (L27-28P3).**
- **We have used the Copernicus Publications LaTeX Package to prepare our manuscript. Semicolons are included when a list of references is provided.**

22. L4-5P3: I do not agree that RCM simulation is cheaper than GCM in terms of computing resources. First RCM needs GCM outputs for boundary model's inputs, second the computing resources depends on the resolutions of the model.

We would like to thank you for pointing this out. We will remove this phrase from the text.

**Modification:**

**The phrase has been removed from the text.**

23. L13-14P3: Unclear sentence. What kind of tools and what will be improved?

We will rephrase the sentence as "This will enable the development of drought prediction and management tools and the improvement of the existing ones".

**Modification:**

**The text has been modified: "A better understanding of the different types of drought and their propagation processes is key to improving current and future drought representation and thus drought prediction and management tools" (L18-19P2).**

24. L15P3: You may remove ","

We will remove the comma from the first sentence.

**Modification:**

**The phrase has been removed from the text.**

25. L15P3: you may replace "capacity" with "capability"

According to the Oxford dictionary, both capacity and capability can be defined as "the ability or power to do something."

**Modification:**

**The text has been rewritten and neither "capacity" nor "capability" appear in it.**

26. L17P3: Too many "it" can you define what are it?

We are sorry that the sentence is not clear. If the "it" are removed, the sentence is as follows: "The comparison allows to analyse the RCM's contribution with respect to the global model that drives the RCM, and see whether if the RCM improves drought simulation". We will rephrase it to make it clearer.

**Modification:**

**The text corresponding to the objectives of the study has been rewritten (L31P3-L3P4).**

27. L19P3: Should be analyzed since they did the studies in the past.

In comments 27 to 34, you suggest to write the text about the state-of-the-art using the past tense. We will follow your advice.

**Modification:**

**The text has been written using the past tense.**

28. L20P3: Replace "uses" with "used" and also define those three drought indices. What are they?
Same reply as comment number 27.

29. L21P3: Replace use(s) with used. The word "too" is informal.
Same reply as comment number 27.

30. L23P3: Replace analyse with analysed.
Same reply as comment number 27.

31. L25P3: Replace use and study with their past tense form.
Same reply as comment number 27.

32. L26P3: you may replace: "..studies are. . ." with ". . .those studies were. . ."
Same reply as comment number 27.

33. L27P3: Replace analyse with analysed.
Same reply as comment number 27.

34. L28P3: Replace are with were.
Same reply as comment number 27.

35. L29-32P3: Why don't you combine these two sentences?
In our opinion, the sentences should not be combined. The first sentence provides results about the study performed by García-Valdecasas Ojeda et al. (2017), while the second one informs about other studies that carry out similar analysis, but do not necessarily obtain the same results. However, since the expression "following this line" was not clear, we will rephrase the second sentence as: "Other studies analysing the effect of the time accumulated period are Vicente-Serrano and López-Moreno (2005), Vicente-Serrano (2006), and Edossa et al. (2009)".
**Modification:**
**Section 1 (Introduction) has been rewritten and the information given in these sentences has been extended (L23-28P3).**

36. L6P4: Missing "the". The standardized indices.
In our opinion, there is no need to include "the" before "standardized indices".
**Modification:**
**Once the text was rewritten, the Scientific Editing Service did not include "the" before "standardized indices".**

37. L7P4: You may replace "with" with "using"
We will replace "with" by "using".
**Modification:**

**The text has been rewritten (L2P4).**

38. L8-10P4: Please write the full meaning of SSMI, SSI, and SRI first since this is the first time you introduce these acronyms.
We will include the acronyms' description in the Introduction section.
**Modification:**
**The meaning of SSMI, SSI, and SRI has been included in Sect. 4 (Methodology), since it is the first time that these acronyms are introduced (L11-13P9).**

39. L11P4: Rewrite: ". . .we will detail the study are".
The sentence will be rewritten as "The study area is detailed in the next section".
**Modification:**
**This paragraph has been deleted.**

40. L18-25P4: This paragraph should be in the result section.
Same reply as the one from "General comments", number 2.
**Modification:**
**The text about Fig. 2 has been moved to the Sect. 5 (Results) (L18-25P11).**

41. L26-27P4: Can you elaborate more why soil moisture anomaly in spring may influence droughts and heatwaves? Also this statement is not belonging to study area.
We agree with you that this sentence may not provide relevant information for the work presented in the manuscript. Therefore, we will rephrase it to explain that soil moisture deficits in Spring over Southern Europe favours drought propagation over Northern Europe. We believe that this information helps characterizing soil moisture drought over Spain and warns about its effect regarding drought propagation towards Northern Europe.
**Modification:**
**This sentence has been rephrased and moved to the first paragraph of Sect. 2 (Area and study period) (L13-15P4).**

42. L29P3: I suggest you to use different word for relief such as mountainous areas?
We believe that this comment refers to page 4. In our opinion, "relief" is a valid word, as it refers to the difference in height from the surrounding terrain, but we can change it if necessary.
**Modification:**
**The word "relief" has not been replaced by another one. However, it can be modified if the Editor and referees deem it necessary.**

43. L31P4: What is "it" refers to?
It refers to "a wide network of dams and river canals" that appears in the previous sentence. We will replace "It" by "This network".
**Modification:**

**This sentence has been removed from the text.**

44. L32P4: Rewrite "to use" into "to be used"
We will replace "to use" by "to be used".
**Modification:**
**This sentence has been removed from the text.**

45. L2P5: You may write: "..a reduction of precipitation. . ."
We will rephrase the sentence as: "However, a reduction of precipitation has been observed  in Spring and Summer (de Luis et al., 2010), as well as an increase in the number of consecutive dry days (Turco and Llasat, 2011)."
**Modification:**
**This sentence has been rephrased as " Although precipitation does not show significant annual trends, observations show a reduction in spring and summer (de Luis et al., 2010) as well as an increase in the number of consecutive dry days (Turco and Llasat, 2011)" (L25-27P4).**

46. L5P5: What is "it" refers to?
It refers to "a rise in annual and seasonal temperatures" that appears in the previous sentence. We will replace "It" by "This rise".
**Modifications**
**"It" has been replaced by "This rise in temperature" (L28-29P4).**

47. L8P5: You write: ". . .increase this type of drought". In my opinion snow melt can increase or decrease the hydrological drought depend on early or late snow melt.
In our opinion the text is correct, as we refer to the "advance of the thaw date" and, therefore, to early snow melt. However, we will include a sentence referring to the fact that streamflows may be affected by snow melt in different ways depending on the snow melt timing and resulting in an increase or decrease of hydrological drought. For this, we will cite Van Loon (2015).
**Modification:**
**The phrase " However, it should be noted that snow melt can affect streamflow and thus hydrological drought in different ways depending on its timing (Van Loon et al., 2010)." has been added (L32-33P4).**

48. L17P5: Why don't you combine paragraph 2 with 1?
Paragraphs from lines 15 to 24 will be combined.
**Modification:**
**The paragraphs describing SAFRAN have been combined (L5-15P6).**

49. L23-24P5: This paragraph stands alone and can be combined with previous paragraph.
Same reply as comment number 48.

50. L28P5: Rewrite: ". . .detailed in Quintana-Segui et al. (2016).

We will rephrase the sentence as: "In Spain, SAFRAN was extended for a 35 year period (1979/1980 – 2013/2014) (Quintana-Seguí et al., 2016)."

**Modification:**

**The sentence has been rephrased: " In Spain, SAFRAN was extended for a 35-year period (1979/1980 – 2013/2014) (Quintana-Segui et al., 2017) and was implemented and validated over the Ebro Basin (north-eastern Spain) (Quintana-Segui et al. 2016)" (L13-15P6).**

51. L31P5: You may replace "in" with "for"

We will replace "in" by "used for".

**Modification:**

**The phrase has been removed the text.**

52. L6P6: Again you may combine all paragraphs in this section (3.1.2) into 1. Paragraph 1 consists of two sentences, which are about ERA-interim and then paragraph two starts with the word "it". You should combine this paragraph since you still discuss about the same thing. Also you cannot start new paragraph with the word "it".

Paragraphs from lines 4 to 9 will be combined. However, paragraphs from lines 10 to 14 will be left alone, since the first one does not describe ERA-Interim, as the previous ones do, and the last one informs about how ERA-interim will be referred to in the rest of the paper. In our opinion, this information should be given in separate paragraphs.

**Modification:**

**The first two paragraphs describing ERA-Interim have been combined (L21-27P5).**

53. L9P6: What is "it" refers to? Spatial resolution?

It refers to "ERA-Interim". This phrase will be moved to line 4 (page6).

**Modification:**

**This phrase has been moved (L23P5).**

54. L11-12P6: please give reference.

We will rewrite the text to better include the Ngo-Duc et al. (2005) and Weedon et al. (2011) references.

**Modification:**

**The text has been rewritten as "However, biases in this type of forcing have a negative effect on LSM simulations, which can be corrected (Ngo-Duc et al., 2005; Weedon et al., 2011)" (L29-30P5).**

55. L12-13P6: You write: ". . .they can be corrected". How?

In our opinion that is out of the scope of this study. In addition, the Ngo-Duc et al. (2005) and Weedon et al. (2011) references, that deal with this issue, are provided.

**This information is not included in the manuscript as it is out of the scope of the study. The references Ngo-Duc et al. (2005) and Weedon et al. (2011), which deal with this issue are given.**

56. L1P7: Rewrite this sentence: "Since there is no observed truth for soil moisture available,. . ."

The sentence will be rephrased as "Since there is no ground truth for soil moisture available, LSM offline simulations are used as reference in this study to analyse both soil moisture and hydrological droughts".

**Modification:**

**The phrase has been rewritten as "Offline LSM simulations are used in this study as a reference to analyse soil moisture drought because there is a lack of soil moisture data across mainland Spain that are suitable for studies that require a large spatial coverage" (L6-7P7). We have excluded "hydrological drought" because offline LSM simulations have been excluded from the hydrological drought analysis.**

57. L8-9P7: You should write the full names first before the acronyms.

The full names will be written before the acronyms.

**Modification:**

**The full names have been written before the acronyms (L3-11P4).**

58. L14P7: You may add "in" before 1 km resolution.

We agree with you that a preposition should be added before "1 km resolution". However, we think that "at" is better than "in".

**Modification:**

**The text has been rewritten as "...derived from the ECOCLIMAP land cover database at a 1 km resolution …" (L17P7).**

59. L14P8: Rewrite the opening sentence.

In our opinion the sentence is correct. However, we will move the reference of Ruti et al. (2015) to the end of the phrase. Regarding this paragraph, we propose to remove the last sentence. In our opinion, it is not necessary to provide further information about Med-CORDEX taking into account that references [Giorgi et al.(2009) and Ruti et al. (2015)] are provided.

**Modification:**

**The opening sentence has been rewritten as "In this study, drought representation and propagation in three RCMs are analysed" (L4P5).**

60. L18P8: Replace "is" with "are" since you mention about drought representation and propagation.

We will replace "is" by "are".

**Modification:**

**"Is" has been replaced by "are" (L4P5).**

61. L20P8: You may reverse the words into ". . .represent physical processes in different ways. . ."
The sentence will be rephrased as : "Each one of them uses a different surface scheme and, therefore, represent physical processes related to precipitation, soil moisture, and surface and sub-surface runoff in different ways".
**Modification:**
**The text has been rephrased as "...each one used a different surface scheme and therefore represented physical processes related to precipitation, soil moisture, and surface and sub-surface runoff in different ways" (L7-9P5).**

62. L22P8: You used the name CNRM-RCSM4 but for previous sentence you use the name RCSM4 only. Please be consistent. Also for COSMO-CLM in L27.
"CNRM-RCSM4" and "COSMO-CLM" will be introduced in Sec. 1, where we will explain that they are known as "RCSM4" and "CCLM4".
**Modification:**
**The models are introduced in Sect. 1 (L3-6P4).**

63. L1P9: In the section 3.4, please mention the number stations that you used in your study.
A total of 87 stations containing at least 95% of data in the study period were selected. This information will be included in the "Observations" section from Sec. 3.
**Modification:**
**This information is included in Sect. 3.2.3 (L22-24P8).**

64. L2-3P9: Missing verb in this sentence.
The sentence contains the following verbs "belong" and "provide".
**Modification:**
**The text has been rewritten as: "Daily streamflow observations were also used as a reference in the hydrological drought analysis. These belong to the Spanish Ministry for the Ecological Transition, which provides data for basins comprising more than one region" (L3-4P9).**

65. L3P9: You may write: "To obtain monthly time series. . ."
We will write "time series", instead of "series".
**Modification:**
**We have written "monthly series" instead of "series" (L11P8).**

66. L4P9: Can you elaborate why south is not as represented as the north?
We filtered the data in order to keep those stations with at least 95% of the data for the period studied and the stations kept are more abundant in the north. This could be due to i) southern basin authorities not submitting all data to the Ministry database, ii) that

stations were installed later and thus the earlier part of the period is not covered, iii) problems of maintenance, iv) a combination of the three. We did not study the causes of this result.

**Modification:**

**The text referring to the south of the peninsula not being as represented as the north has been removed.**

67. L5P9: You may rewrite "to having" with "to have"

If "to having" is replaced by "to have", we will change the sentence's meaning. What we intend to say is that we can either have i) more stations containing less data or ii) less stations containing more data. We decided to work with less stations with the certainty that these have at least 95% of data in the study period. In our opinion the sentence is correct. However, we can rephrase it to make it more clear.

**Modification:**

**This phrase has been removed from the text.**

68. L7-8P9: You wrote: "In this way, both studies complement each other. . .." It is not clear sentence. What is in this way?

It's equivalent to saying "Like this". The same methodology is used in the study performed by Quintana-Seguí et al. (2019) and the work presented in this paper. Therefore, both studies are complementary.

**Modification:**

**"In this way" has been replaced by "Therefore" (L7P9).**

69. L10-11P9: This sentence is not clear. You may write: ". . .we compute SPI and SSMI, which require precipitation and soil moisture data, respectively".

We propose to rephrase the sentence as: "For the meteorological and soil moisture drought analyses, we compute SPI and SSMI using precipitation and soil moisture data, respectively".

**Modification:**

**The text has been rewritten as "For the meteorological and soil moisture drought analyses, we computed the SPI and a Standardized Soil Moisture Index (SSMI) using precipitation and soil moisture data, respectively" (L10-12P9).**

70. L14-16P9: Proper citations for all indices you mention here, except if you already did so before.

In the text it is said that "We follow the spirit of the Standardized Precipitation Index (SPI)..." (L14P9). Therefore, apart from the SPI index computed using precipitation, other standardized indices are computed using other variables and following the same methodology as the one followed to compute the SPI (referenced in L6-7P3). The text provides these variables and their corresponding indices. We propose to increase the references to other studies that employ standardized indices computed using other variables besides precipitation.

**Modification:**
**References about studies that employ standardized indices using precipitation, soil moisture, runoff, and streamflow are given in the Introduction section (L33P2-L6P3).**

71. L1P10: You said the variable's time series is transformed from its original distribution to a normal one. What kind of distribution? Gamma, GEV, Pearson, etc?.
There are several parametric distribution functions used to compute standardized indices. We propose to rephrase the sentence, since it may not clear if the text refers to standardized indices in general or to the indices computed for the study described in the manuscript.
**The reference to (Farahmand and AghaKouchak, 2015) is included in the text: " To solve this problem, we used the non-parametric methodology described by Farahmand and AghaKouchak (2015)" (L8-9P10).**

72. L3P10: You may rewrite: ". . .in the meteorological case. . ." into "for meteorological case. . .."
The text will be rewritten.
**Modification:**
**The text has been rewritten as "The computation was carried out using monthly data for all indices. In the case of the SPI, a time series of the accumulated precipitation from the previous *n* months (with *n* being the index scale) was also calculated to perform the drought propagation analyses " (L4-6P10).**

73. L4P10: You may change "for" with "from" and also missing verb.
We will replace "for" by "from". The sentence contains the verb "to use" in its -ing form. However, we will rewrite the sentence as: "The computation is carried out using i) monthly data for all indices and ii) a time series of the accumulated precipitation from the previous n months (being n the index scale) for the SPI".
**Modification:**
**We have replaced "for" by "from".The text has been rewritten (see previous comment).**

74. L8P10: Why only for SPI-12?
Considering 12 months results more robust due to seasonal reasons. We will mention this in the text.
**Modification:**
**The following phrase has been included in Sec. 4.1: " Using results from an accumulation period of 12 months is particularly robust due to seasonal reasons" (L11P10).**

75. L13-14P10: Please give reference.
The reference McKee et al. (1993) will be given.

**Modification:**
**The reference McKee et al. (1993) is given (L17P10).**

76. L17P10: Please use better opening sentence.
We will rephrase the sentence as "The methodology employed to analyse meteorological drought propagation is based on Barker et al. (2015)".
**Modification:**
**The text has been rewritten as "The analysis of how meteorological drought propagates to soil moisture and hydrological drought is useful in hydrological resource management and allows the detection of similarities and differences in the way models address the physical processes that drive this propagation. The methodology employed to analyse meteorological drought propagation is based on Barker et al. (2016)" (L20-23P10).**

77. L19P10: You may write actual evapotranspiration.
We will remove "potential" from the text.
**Modification:**
**The word "potential" has been replaced by "actual"  (L25P10).**

78. L15P11: You can combine second paragraph with the first.
Paragraphs from lines 13 to 19 will be combined.
**Modification:**
**Both paragraphs have been combined (L16-25P11).**

79. L17-18P11: What do you mean with relief is a determining factor in distribution?
The text aims to indicate that the way water from precipitation is distributed over a region is strongly influenced by the region's relief. We will modify the text to refer to water distribution from precipitation.
**Modification:**
**The text has been rewritten as "The fact that precipitation is high over mountainous chains indicates the strong influence of relief, which is key in the way water from precipitation is distributed" (L22-23P11).**

80. L21P11: Modeled precipitations usually have higher results than the observed and ERA-i. There are some papers show that.
References will be included.
**Modification:**
**The reference Sylla et al. (2010) has been included (L27-28P11).**

81. L4P12: Drought in 2005-2006 does not coincide with those found "in" Belo-Pereira et al. (2011) but it occurred. So please rephrase.

The paper from Belo-Pereira et al. (2011) explains that "In 2005–2006, the available data sets (CRU, GPCC and ERA-I) agree on a generalized drought spell in all areas". Therefore, drought in 2005-2006 is observed by the CRU, GPCC, and ERA-I data sets. In our work, the three RCMs analysed, as well as the ERA-I (RCMs' driving data) and the SLR (reference data) data sets also show a drought in 2005-2006.

**The text has not been modified because Belo-Pereira et al. (2011) showed that the datasets CRU, GPCC and ERA-I agree on a generalized drought spell in 2005–2006 and the RCMs analysed in the study we performed identify this spell too.**

82. L6P12: It is hardly to see in the Figure. Can you write the number of months?

The number of months will be provided.

**Modification:**

**The text has been extended and the drought spell's number of months have been included (L4-7P13).**

83. L2P13: typo for tan

We will correct the typographical error.

**Modification:**

**The typographical error has been corrected.**

84. L10P13: You may write: "The difference is the degree to which they deviate"

We will replace "The difference is the degree to which they do it" by "The difference is the degree to which they deviate".

**Modification:**

**The phrase has been rewritten as "The difference relates to the degree to which they deviate" (L31P13).**

85. L11P13: What are "it" refer to?

The word "it" refers to the percentage of area affected by drought. The text will be rewritten as: "On the one hand, in 1995, RS4 overestimates the percentage of area affected by drought by 20 %, CL4 around 15 % and PMS does not reach 10 %. On the other hand, in 1996, PMS underestimates this percentage by more than 30 %, while RS4 and CL4 underestimate it by 10 %".

**Modification:**

**The phrase has been rewritten as "On the one hand, in 1995, RS4 overestimates the percentage of area affected by drought by 20%, CL4 by approximately 15%, and PMS by less than 10%. On the other hand, in 1996, PMS underestimates this percentage by more than 30%, while RS4 and CL4 underestimate it by 10%" (L31-34P13).**

86. L12P13: What do you mean with "do it"

The words "do it" refer to the action of underestimating the percentage of area affected by drought. The text will be rewritten to make it clear (see the previous comment).

**Modification:**
**See the modification from comment number 84.**

87. L6P15: How about the white color in the table?

Regions whose values are not within the colour scale are represented in white. We will inform this in the text.

**Modification:**
**We have informed in the captions of Figs. 5 (P16) and 6 (P18) that regions whose values are not within the colour scale are represented in white.**

88. L12P15: Please rewrite: ". . .RCMs compared between them. . ."

We are not sure about the reason why it is proposed to rewrite this sentence. Our purpose is to show that if we compare drought representation by RCMs with other RCMs, we see changes in drought category representation. This is expected because, as explained in the text's previous phrase, changes in drought category representation are identified when RCMs are compared to LSMs (the reference data sets). But we can rephrase the text to make it more clear.

**Modification:**
**The text has been rewritten as "In addition, the three RCMs compared among them also represent soil moisture droughts of different categories (second and third columns), as expected" (L8-9P16)**

89. L14P16: Please elaborate more why SPI with higher accumulation period is slower.

Fig. 5 shows the month scale at which the correlation between SSMI-1 (SSMI index computed for a time accumulation of 1 month) and SPI-n (SPI index computed for a time accumulation of 1 to 28 months) is maximum. Therefore, it can be interpreted as the temporal scale at which meteorological drought propagates to soil moisture drought. As the number of months increases, the models' dynamics decreases. Bearing this in mind, Fig. 5 does not show that the SPI is slower with higher accumulation periods, but that in the context of meteorological drought propagation to soil moisture drought, some models show slower dynamics (ORCHIDEE) than others (ISBA).

**We believe that the text from Sec. 5.2.2 (Propagation to soil moisture drought) together with Fig. 6 explain in a proper way the results obtained for the meteorological drought propagation to soil moisture drought. In addition, the methodology used is detailed in Sec. 4.2 (Meteorological drought propagation).**

90. L3P17: you may rewrite: ". . .to the LSM with the surface scheme it is coupled to." Into ". . .LSM coupled with the surface scheme."

We propose to rephrase the sentence as: "This becomes more evident, when an RCM is compared to the LSM that has the same surface scheme".

**Modification:**
**The text has been rewritten as "This becomes more evident when a RCM is compared to the LSM that has the same surface scheme" (L7-8P17).**

91. L7P18: Rewrite: "To single out stations"

In our opinion, the text "To single out stations" is correct. According to the Oxford Dictionary, to single someone/something out is defined as "Choose someone or something from a group for special treatment", which corresponds to the actions described in the text.

**Modification:**

**This information has been moved to Sect. 3.2.3 (P8). The text has been rewritten, and the expression "To single out stations" has been removed.**

92. L19-20P18: I thought LSMs should have better results for hydrological drought than GCMs.

In the first place, it should be noted that hydrological drought is not discussed in this subsection. In our opinion, the ideal variable to analyse hydrological drought is streamflow. Since RCMs do not provide simulated streamflow, we propose to use aggregated runoff to compute a standardized index (SRI) to study hydrological drought. For this, we first compare streamflow and aggregated runoff between them to analyse their resemblance by means of the Kling-Gupta Efficiency (KGE). This subsection addresses the comparison between both variables (streamflow and runoff). In the second place, we would like to point out that there is no comparison between LSMs and GCMs, the comparison is between RCM simulations driven by ERA-Interim and offline LSM simulations forced by i) ERA-Interim and ii) SAFRAN. RCMs provide better KGE values than LSMs, which means that aggregated runoff simulated by RCMs performs better than aggregated runoff simulated by LSMs when compared to streamflow and when forced by the same forcing data (ERA-Interim).

**The aim of this subsection (Sec. 5.3.1 Streamflow and aggregated runoff comparison) should be clearer now.**

- **In Sec. 5.3 (Hydrological drought) it is said that in Sec. 5.3.1, streamflow and aggregated runoff will be compared.**
- **The results showing the comparison between RCMs and offline LSM simulations (used as reference datasets) have been deleted to avoid misunderstandings. Therefore, the columns "ERA-ORC", "ERA-ISB", and SFR-ISB" have been removed from Table 4 together with the text referred to these columns.**
- **Sec. 3 has been restructured to explain in a clearer way the datasets used as reference (for example, offline LSM simulations).**

93. L22P18: I saw 0.6 and not 0.7. Also you said that PMS behaves better over both basins. How do you define PMS is better? By average value or by color?

The KGE value at station number 9002 for the comparison between RS4 and observations (lower table) is 0.7. The value 0.6 corresponds to the comparison between RS4 and the SIMPA model at the same station. We will modify the text to specify that 0.7 corresponds to the comparison between RS4 and observations.

We have identified PMS as the best performing according to the average value. We will include this in the text.

**Modification:**

**The text has been rewritten as "Although RS4 is the RCM showing the highest KGE value (0.7 at station number 9002 for the comparison between RS4 and OBS), PMS performs better across both basins according to the average value" (L14P18-L2P19).**

94. L4P19: Again what is white?

Regions whose values are not within the colour scale are represented in white. We will inform this in the text.

**Modification:**

**The text has been rewritten as "Negative values, indicating poor performance, are marked in red. For the positive KGE values, a colour scale consisting of a blue (best performance between RCMs and i) SMP or ii) OBS) to red (worst performance between RCMs and i) SMP or ii) OBS) gradient via white has been included to facilitate reading" (L9-12P18).**

95. L14P21: You may change from "the 9025 station" to "station number 9025"

We will replace "the 9025 station" by "station number 9025".

**Modification:**

**The text has been rewritten as "... reference data in station number 9025 (which represents the Segre sub-basin)" (L6-7P19).**

96. L10P22: What is extension?

Extension refers to the area affected by a drought event. We will replace "extension" by "area" to make the text more clear.

**Modification:**

**The word "extension" has been replaced by "area" (L3P22).**

97. L33P23: You may rewrite: "..that PMS' precipitation extremes are too strong" into ". . .that PMS simulates higher precipitation amount."

In our opinion, the phrase is valid. The paper provided as a reference (Dominguez et al., 2013) studies extreme precipitation indices. Therefore, the manuscript's text should refer to extreme precipitation.

**The text has not been modified because the reference provided deals with extreme precipitation indices.**

98. L9P24: I believe you cannot avoid the error but you can minimize the error.

We will replace "avoiding" by "minimizing".

**Modification:**

**The word "avoiding" has been replaced by "minimizing" (L6P24).**

99. L15P24: Replace use with used. Past tense passive sentence.
   We will replace "use" by "used".
   **Modification:**
   **The word "use" has been replaced by "used" (L12P24).**

100.   Figures: They are too small, cannot see them clearly.
   Figures will be enlarged.
   **Modification:**
   **Figures and tables have been enlarged. However, there are axis and scale levels that might still be too small and might have to be enlarged more.**

101.   Table 6: I cannot find the number in bold. Can you also please write the correlation numbers for each color in the caption?
   We would like to thank you for pointing this out. Scales longer than 12 months will be marked in bold and the correlation ranges for each colour will be included in the caption.
   **Modification:**
   **Scales longer than 12 months have been marked in bold and the correlation ranges have been included in the caption (P21).**

We hope you will see a clear improvement in a revised version of the manuscript.

Yours sincerely,

Anaïs Barella-Ortiz

**Referee 2**

Dear Referee,

Thank you for taking the time to read the manuscript, as well as for commenting on parts of it. Below you will find the answers to the comments you made about the manuscript titled "Evaluation of drought representation and propagation in Regional Climate Model simulations over Spain" by A. Barella-Ortiz and P. Quintana-Seguí.

**General comments**

1.   The language throughout makes it difficult to understand at times (for example, P23L18-19: It is not clear if this means that RCMs are not appropriate to calculate SPI at longer accumulation periods) and should be improved dramatically before publication. The manuscript will be sent to a Scientific Editing Service to assure that English is correct. Our purpose in P23L18-19, is to point out that the time accumulation period used to compute standardized indices influences the indices' added value. Therefore, we do not state that RCMs should not be used to compute SPI. However, the text will be rewritten to clarify this idea.
     **Modification:**
     ● **The manuscript has been sent to a Scientific Editing Service.**
     ● **The text mentioned in this comment has been simplified: "According to GarcÍa Valdecacas Ojeda et al. (2017) and Bowden et al. (2016), the added value of the SPI is influenced by the accumulation period employed. This is interesting to consider because soil moisture and hydrological droughts have different scales of propagation, and differences are also found in the propagation analysis in Sect. 5.2.2 and 5.3.3". (L11-14P23)**

2.   The paper lacks clearly defined aims and applications, currently it reads as a modelling exercise rather than science to support real-world applications – this can be addressed by a better structured introduction as it currently jumps around without a coherent story (e.g. what is the problem, what have others done in the past, what is the research gap, what is the aim of this study and how this will address the research gap).The introduction and literature review also relies heavily on the IPCC reference, without reviewing peer reviewed publications (and where papers are introduced, they are often listed as 'other papers on the topic' such as P2 L32) and outlining the research gap this paper is aiming to fill. The lack of aims and disjointed nature of the paper make it difficult to reach the conclusions set out in the final section of the paper.
     Drought is a climatic risk, which will become more frequent, severe, and lasting due to a warmer climate. Therefore, it is important to know the evolution of drought. For this, the current modelling tools must be first evaluated. There are several studies about drought representation by models using different types of drought indexes. Our aim is to

contribute to these studies by analysing how regional climate models represent drought, as well as the propagation from a precipitation anomaly to a soil moisture and streamflow anomaly. In addition, our study complements a previous one by Quintana-Seguí et al. (2019) which analyses drought representation and propagation by land-surface models using the same methodology as we have employed. As a result, this study improves our current knowledge on the regional climate models' capability to reproduce both drought and its evolution.

The Introduction section will be rewritten and clarified. We will provide and explain more references to avoid lists. In addition, to increase the focus of the paper on the analysis of the RCMs, we propose to delete results from LSM simulations in the hydrological drought analysis (Tables 4 to 6: columns "ERA-ORC", "ERA-ISB", and "SFR-ISB"), together with the text referring to these.

**Modification:**

- **Section 1 (Introduction) has been rewritten.**
- **Section 3 (Datasets) has been restructured to clarify that RCMs are the models analysed and SAFRAN, offline LSM simulations, a SIMPA simulation, and observed streamflow are used as reference data.**
- **The results referred to LSMs shown in Section 5.3 (Hydrological drought) have been removed to avoid misunderstandings.**
- **The manuscript has been sent to a Scientific Editing Service.**

3. It has not been made clear why this modelling approach was used. The assertion on P6L20 that the atmospheric feedback not being accounted for makes LSMs a good tool to study drought because they can be treated like physically distributed hydrological models, does not provide explanation – why do you want them to behave like a physically distributed model?

The reasons for using LSM offline simulations as reference in the soil moisture drought analysis, are the following:

1. Using offline simulations avoids biases due to the atmospheric model and the coupling between LSM and atmospheric model.
2. There is no observed truth available for soil moisture (P7,L1-2) that covers mainland Spain, thus we use offline simulations as a reference.

We will rewrite the text in P6L20 to explain clearer that atmospheric model biases are excluded in LSM offline simulations.

**Modification:**

**The text has been rewritten as "Offline LSM simulations are used in this study as a reference to analyse soil moisture drought because there is a lack of soil moisture data across mainland Spain that are suitable for studies that require a large spatial coverage. Internally, in RCMs, LSMs are bidirectionally coupled to the atmospheric model to simulate surface processes. However, when LSMs are run offline (forced by gridded databases), biases due to the atmospheric model and the coupling between the atmospheric model and the LSM are avoided. This**

makes offline LSM simulations good reference datasets to study drought"
(L6-10P7).

4. You don't mention or address the issue of uncertainty – what about the uncertainties of the modelling approach? Could you explore this using a multi-model ensemble?
Yes, uncertainty is a very important topic and, in fact, we are already dealing with it. The main objective of this article is to evaluate drought properties in RCM simulations. We are performing the analysis on three different models and these provide different results in terms of drought indices and, specially, drought propagation. With only three models we already show that the values of $n_x$ (time scale of propagation of drought) are very different for both soil moisture and runoff. Adding more models would show the spread with more detail, but we believe that the differences with three models are large enough to show that RCM developers should look at these issues and RCM users should take them into account.
**Modification:**
**The issue of uncertainty has been included in Section 6 (Discussion) (L18-22P23).**

5. In many places, there is text seemingly in the wrong section of the paper, for example P3L33-P4L4 should more likely sit in the data/methods section as this is detailed for the introduction. P5L1-8 would be better placed in the introduction. P11L24-27 would be better placed in the discussion describing why there were discrepancies between the modelled outputs. The authors should review the text to ensure that descriptions of data and methods and discussion text are in the appropriate sections.
   - P3L33-P4L4 should more likely sit in the data/methods section as this is detailed for the introduction. In our opinion it is not too detailed for the introduction as we list the RCMs analyzed and the references used. These are further explained in Section 3. However, if the referees and editor deem it necessary, this part of the text from the Introduction section can be shortened.
       - **The RCMs and the reference data used are listed in Sect. 1 (Introduction). All of them are explained in Sect. 3 (Methodology), the RCMs are detailed in Sec. 3.1 (Regional Climate Models) and the reference data in Sec. 3.2 (Reference products).**
   - P5L1-8 would be better placed in the introduction. The text describes drought in mainland Spain, which is the area of study. That is the reason why it is located in Section 2 (Area and study period). However, it can be moved to the Introduction Section if the referees and Editor deem it necessary.
   **Sect. 2 has been kept. However, some of the text has been moved to Sect. 5.1.1 (Mean annual precipitation comparison).**
   - P11L24-27 would be better placed in the discussion describing why there were discrepancies between the modelled outputs.
   We propose to move the phrases " There are few pluviometers in mountainous areas, probably causing an underestimation of precipitation over these regions. However, this effect is limited to mountainous areas" to the Discussion Section.

However, we believe that lines 26 and 27 should remain in Section 5.1.1, because they describe CCLM4 mean precipitation over mainland Spain.

**Modification:**

**The phrases "There are few pluviometers in mountainous areas, probably causing an underestimation of precipitation over these regions. However, this effect is limited to mountainous areas." have been removed from the text.**

6. Section 3.1: this should have more introductory information before diving into the detailed descriptions of SAFRAN and ERA, in 3.1 please outline what variables you use and what they are needed for before describing them in turn.

More introductory information will be provided before the subsections describing SAFRAN and ERA-Interim. Besides referring to Table 1, we will explain why these were chosen as driving data and reference dataset.

The variables used in our study to compute standardized indices (precipitation, soil moisture, streamflow, and total runoff), are detailed in Section 4.1 (Drought indices calculation) (P9L14-16), not in Section 3.1. However, we do provide a description of SAFRAN's precipitation in Section 3.1.1 because it is considered our reference dataset for the meteorological drought analysis.

**Modification:**

**The variables used to compute the standardized indices are provided in the Sect. 1(Introduction) as well as for which drought analysis they are used.**

**Sect. 3.1 has been modified to make the manuscript clearer:**

- **Since ERA-Interim is the driving data of the three RCMs analysed in this study, it is described in Sect. 3.1 (Regional Climate Models) (L21-32P5).**
- **SAFRAN is described in Sect. 3.2.1 (Meteorological drought reference) (P6). In the description we explain why it is chosen as the reference dataset for the meteorological drought analysis. Although, all the variables estimated by SAFRAN are provided, focus is made on precipitation, since it is the variable used to compute the SPI.**

7. In Section 3 the RCMs are introduced third but surely start the modelling chain, I suggest you introduce these first, then the LSMs then Hydrological Models. Was it necessary to calibrate and validate your models – how did you do this?

Yes, we can clarify this point. You mention "modelling chain" as if the RCMs were driving the LSMs (one way forcing). We fear there is a confusion here. The RCMs contain themselves a LSM, which is coupled with the atmospheric model of the RCM. We are taking the outputs of the RCM's LSM variables directly from the MedCordex database. Thus, there is no "modeling chain" in this regard. This is, we are not forcing standalone LSMs with the outputs of the atmospheric variables of an RCM simulation. We also use standalone offline LSM simulations, forced by ERA-Interim and SAFRAN, in order to have comparison points.

That being said, we propose to restructure Sect. 3 as follows, in order to provide more clarity:

    3 Datasets
        3.1 Forcings and driving data
            3.1.1 SAFRAN meteorological analysis
            3.1.2 ERA-Interim
        3.2 Models
            3.2.1 Regional climate models
            3.2.2 Land Surface Models
                SURFEX
                ORCHIDEE
            3.2.3 Hydrological models
        3.3 Observations

SURFEX and ORCHIDEE sub-sections will not be numbered, because only three levels of sectioning are allowed.

Concerning model calibration, the situation is as follows:

1. We took RCM data from the MedCordex database. RCM modelers do tune their models, but this information is not available to us. In this regard, we simply are users of the RCM outputs.

2. We used an ERA-ORCHIDEE simulation, which was provided to use by Jan Polcher (IPSL). We do not know how IPSL calibrates its LSM.

3. We performed ERA-SURFEX and SAFRAN-SURFEX simulations. We did not calibrate SURFEX. We used default values for all non physical variables (i.e. subgrid runoff). The corresponding flows were calculated using the RAPID river routing scheme. The Muskingum parameters were not tuned, we used default values too. Concerning SURFEX modelled flows, we have proposed in comment number 2 to delete results from LSM simulations in the hydrological drought analysis.

4. We also used the outputs of the SIMPA model, as a reference. This model is heavily calibrated. The model calibration and run were performed by CEDEX, the Spanish institute that provides the reference natural streamflow simulations to the Ministry for the Ecological Transition and the basin authorities. We are users of these simulations and do not have access to information on the calibration.

This information can be included in the manuscript if the referees and editor deem it necessary.

**Modification:**
**The structure of the manuscript has been modified. However, we did not follow the structure proposed previously in this comment. Section 3 (Datasets) has been modified in the following way:**
**3 Datasets**
        **3.1 Regional climate models**

**3.2 Reference products**
**3.2.1 Meteorological drought reference**
**3.2.2 Soil moisture drought reference**
**3.2.3 Hydrological drought reference**

**We believe that this structure clarifies that RCMs have been analysed and SAFRAN, offline LSM simulations, SIMPA, and observed streamflow have been used as reference data. In addition, Tables 1 and 2 have been modified. Table 1 provides the RCMs analysed in the study and Table 2 lists the reference data used in the study.**

8. A lot of detail is provided about the LSMs which is published elsewhere and appears to pad the paper, much of the model background can be removed – the focus should be on why the models were chosen and what they will be used for.

   LSMs' descriptions will be shortened and the reason why they were chosen and the use they were given in the study will be explained.

   **Modification:**

   **The LSMs' description has been shortened. Apart from a brief description, the text explains the following:**

   - **Offline LSM simulations are used as reference datasets in the soil moisture drought analysis (L6P7).**
   - **The offline LSM simulations selected are used as surface schemes by two of the RCMs analysed in this study (L10-11P7).**

9. Section 3.4: It would be useful to include a map of the observation stations used- how many stations were used? Only 8 across the whole of Spain? Why not more – there must be more than 8 stations that have 95% data completeness?

   You are right, there are more than 8 stations with 95% data completeness. However, our criteria to select them was more demanding:

   - 95% data completeness: This assures that the observation monthly series have few gaps.
   - Area greater than $10^3$ km$^2$: Since streamflow is approximated by runoff, it is likely to perform poorly in small basins considering the coarse resolution of the RCM simulations. Therefore, the analysis was limited to large areas.
   - KGE between SIMPA and the observations greater than 0.5: To consider a near-natural regime.

     These criteria are explained in Section 5.3 (P18L1-11). The second one was the most restrictive, since only 13 stations out of the 87 with 95% data completeness have an area greater than $10^3$ km$^2$.

     **Modification:**

     **This information has been moved to Sect. 3.2.3 (L11-21P8), where the reference datasets for the hydrological drought analysis are explained.**

10. P18L8: What evidence or scientific literature did you use to select the 'arbitrary' KGE of 0.5? Later in Section 5.3.1 you say performance of CL4 is poor because KGE is generally below 0.5, but the best performance is for RS4 with KGE of 0.7 – is this enough of a difference between poor and best (reading 'good') performance?

● P18L8: What evidence or scientific literature did you use to select the 'arbitrary' KGE of 0.5?

In order to validate the aggregated runoff of the RCM simulations, we needed gauging stations that were as natural as possible and whose corresponding basins were large enough to be compared to a low resolution RCM. This is difficult in Spain, due to the high degree of human influence. Thus, we needed to have enough large basins that were as natural possible. We thought that a high value of KGE between SIMPA (naturalized flow) and the observations was an indicator of natural regime. Then, we had to set a threshold of the KGE. We tested different values and 0.5, was a reasonable compromise between "near natural regime" and "enough number of stations". It is true that 0.5 is not very high, and thus some human influence can be present, but we had to draw the line somewhere and we did a sensitivity analysis based on our own judgement. Thus, the value is not as "arbitrary" as the text implies. We propose to remove the word "arbitrary" from the text, as it is misleading, and clarify the procedure we followed to select this value.

**Modification:**
**The text has been rewritten (L15-21P8).**

● Later in Section 5.3.1 you say performance of CL4 is poor because KGE is generally below 0.5, but the best performance is for RS4 with KGE of 0.7 – is this enough of a difference between poor and best (reading 'good') performance?

Regarding the difference between poor and best, the text in Section 5.3.1 does not refer to a "best performance":

○ It compares RCMs with LSMs and states that two RCMs provide better KGE than the LSMs with the same surface scheme (P18L19-20).

○ We say that CCLM4 behaves poorly because we have set a minimum threshold of 0.5 and all KGE values provided by the comparison of CCLM4 and SIMPA are lower.

○ When RCSM4 and PROMES are compared between them, it is said that RCSM4 shows the best KGE value for one station, but that PROMES behaves better over both basins (P18L22).

We propose to modify the text so that it is clearer that CCLM4 provides the worst performance of the three RCMs analyzed and to replace " PMS behaves better" by "PMS performs better". In addition, we will explain that PMS is identified as the best performing RCM out of the three RCMs analysed according to the KGE average value. In this section, We will also avoid using adjectives such as "poor" to qualify the KGE values, sticking more to the numerical values, in order to avoid misleading the reader with our own subjective views.

**Modification:**
**The text has been rewritten as " Focusing on the positive KGE values, CL4 has the worst performance, since the values are below 0.5. Although RS4 is the RCM showing the highest KGE value (0.7 at station number 9002 for the comparison between RS4 and OBS), PMS performs better across both basins according to the average value" (L12P18-L2P19).**

11. Section 5.3.1 – why were the temporal analysis not shown? If you only have 8 gauging stations, it would be simple to include time series plots showing the modelled ensemble data against the observations.
    Temporal analysis are not shown in order to reduce the number of figures. The main result of this section is Table 4, which compares streamflow and aggregated runoff to analyse the performance of RCMs by means of the KGE. The temporal series provides additional information, which in our opinion is not necessary to include. However, if the referees and the editor deem it necessary, we can include them.
    **A figure showing the temporal analysis has not been included. As explained in the previous reply, the main result of this section is Table 4.**

12. Table 6: This might be better as a figure with the catchment areas coloured by SPI-nx – as readers we don't know where your catchments are, how do the results vary spatially? What might the effect of catchment properties be on the propagation process? How well do the different models represent these catchment properties?
    We thought about making a similar figure, but we discarded it. The figure we planned to make was to plot the points of the stations, graduating their color in function of the value of $n_x$. However, as we are using only 8 stations (because we want to compare the simulations to observations from near natural flow gauging stations), the maps were not really necessary (a lot of empty space, with just a few colored points). You propose to color the areas of the basins. This can be done, but, again, with only 8 stations, the resulting colour areas would not be meaningful enough. However, we propose to include a figure showing the relief, river network, and the ubication of the gauging stations. We will include, if possible, the catchment areas defined by each station.
    **Modification:**
    **A figure showing the relief, river network, and the location of the gauging stations has been included (Fig. 1, P9).**

13. Table 6: What r values are associated with the Evans classification? How significant are these correlations? The bold type face mentioned in the caption is not obvious in the table.
    - The correlation ranges from the Evans (1996) classification will be included in the caption.
      **Modification:**
      **The correlation ranges have been included in the caption of Table 6 (P21).**

- Correlations are 95% significant. This information will be included in the manuscript.
  **Modification:**
  **The phrase "... and the Pearson correlation coefficient (r) at a 95% significance level." has been included in Sect. 4.1 (L15P10)**
- Scales longer than 12 months will be marked in bold.
  **Modification:**
  **Scales longer than 12 months have been marked in bold in Table 6 (P21).**

14. P22L4: why do you believe standardised indicators are appropriate for this study? This should have been outlined previously.
    Standardized indices define drought according to the variability of a given variable (P4L6 and P9L9-10). They allow to study different types of drought depending on the variable selected. In addition, variability is very important regarding drought analysis and it is the basis of these indices. We propose to extend the text to make clearer why we used standardized indices.
    **Modification:**
    **The text from Sect. 4 has been extended to mention the importance of variability, which is key in standardized indices, in drought analysis (L10P9).**

15. P22L10: what is meant by event extension? The duration and intensity (and extension) of events is not described elsewhere nor shown in any figures – what do you refer to here?
    By "event extension" we refer to the area affected by a drought event. We will rewrite the text to make it clearer.
    The duration and intensity of events is shown in Fig. 2, while the extension (area) is treated in Fig. 3.
    **Modification:**
    **The word "extension" has been replaced by "area" (L3P22). More information about drought duration and intensity has been included in Sect. 5.1.2 (L4-13P13).**

16. In general the figures were too small and labels too small to read. You should avoid the red-blue colour schemes of Tables 3-6, they are not appropriate for those who are colour blind. You can check whether your figures are colourblind friendly here: http://www.color-blindness.com/coblis-colorblindness-simulator/ or by using the CVSimulator app
    We will increase the figures' size. We will also check the tables in the Color BLIndness Simulator and change the colour schemes to make them more appropriate to colour blind people. Thank you for pointing this out.
    **Modification:**
    - **The figures have been enlarged. However, there are axis and scale levels that might still be too small and might have to be enlarged more.**

- **The colour schemes from Tables 3 to 6 has been changed from green-red to blue-red and from green-blue-red-white to blue-yellow-red-white. We hope that these colour schemes are more colourblind friendly.**

17. In regards to the Barker et al. (2015), you have cited the Discussions paper, please cite the final 2016 paper (https://www.hydrol-earth-syst-sci.net/20/2483/2016/hess-20-2483-2016.html).
Thank you for pointing this out. We will cite Barker et al. (2016).
**Modification:**
**Barker et al. (2016) has been cited (L23-24P10).**

18. On P3L31 Lopez-Morreno's name has been misspelled.
Thank you for pointing this out. We will correct the reference.
**Modification:**
**The reference has been corrected (L27P3).**

19. The tense throughout is the present tense, however, research is conventionally written up in the past tense (as it is work that has been completed), please correct this in the next version of the paper.
We propose to write the state-of-the-art from the Introduction section, as well as the Methodology section in the past tense and leave the results description in the present tense.
**Modification:**
**The state-of-the-art from the Introduction section, as well as the Methodology section have been written in the past tense.**

We hope you will see a clear improvement in a revised version of the manuscript.

Yours sincerely,

Anaïs Barella-Ortiz

**Relevant changes made in the manuscript**

- Sect. 1(Introduction) has been rewritten.

- Sect. 3 (Datasets) has been restructured to distinguish the RCMs analysed in the paper from the reference datasets used (SAFRAN, offline LSM simulations, SIMPA simulation, and observed streamflow). The section is divided into two subsections: i) Regional climate models and ii) Reference products. Tables 1 and 2 have also been modified to differentiate RCMs from the reference datasets.

- A figure including the river network, relief and the location of the gauging stations selected in the hydrological drought analysis has been included (Fig. 1).

- The results referred to offline LSM simulations showed in Sect. 5.3 (Hydrological drought) have been removed to avoid possible misunderstandings with the aim of the study.

- The manuscript has been sent to a Scientific Editing Service.

**Marked-up manuscript version**

A manuscript marked-up version is included below. Changes are highlighted in red. A section title that is highlighted in red corresponds to a new section or to a section that has been largely modified.

[revised manuscript text omitted]

---

## Author Response (AR2)

**Point-by-point response to the referee's review and marked-up manuscript version**

The page and line numbers indicated in the replies are referred to the marked-up manuscript version.

Dear Referee,

Thank you very much for taking the time to read the revised version of the manuscript and commenting on parts of it. Below you will find the answers to the comments you made.

1. The abstract is difficult to read. The authors also mentioned the aim of the paper as well as the objective. They can combine these two.
The abstract is divided into four paragraphs:
- Brief introduction explaining why it is important to analyse drought modelling
- The aim of the study. The phrase about the objectives has been removed to make clearer the abstract. This paragraph also informs that the study has been carried out by means of standardized indices.
- RCMs analysed and references used.
- Main results.

2. P2L2: Although the paper does not discuss the socioeconomic drought, it is still worth to mention in this paragraph.
The phrase "The latter case, corresponds to socioeconomic drought and is out of the scope of this paper" has been included in the paragraph (P1L23-P2L1).

3. P2L22: I do not fully agree if RCM is frequently used at the catchment scale, except for big catchment.
The phrase has been rewritten as "Such models are suitable for drought analyses performed at the scale of large river basins" (P2L20-21).

4. P3L8-29: This paragraph can be shortened.
The paragraph has been shortened (P3L6-24).

5. P3L30-31: I suggest to write a better opening sentence in line with the previous one.
The opening sentence has been rewritten as "Taking into account the different types of drought analyses performed with RCMs and the need to better understand droughts at the current time, but also in the future, the regional modelling tools must be evaluated" (P3L25-26).

6. P4L1-2: Since this paper has 2 objectives, so it should be written: These are....
The text "This is a key issue for the understanding …" refers to the second objective of the study. To avoid a possible misunderstanding, we propose to replace "This is a…" by "The latter objective…" (P3L31).

7. P4L15-17: This sentence is unclear.
The sentence has been removed for the sake of clarity.

8. P5L4: Remove the word "analyzed" and add "in the study" at the end of the sentence.

The text has been rewritten as "This section describes the RCMs (Sect. 3.1) and the reference products (Sec. 3.2) used in the study" (P4L30).

9. P6L15: ERA interim.
We believe that the comment refers to P6L14. In our opinion "ERA" should not be replaced by "ERA-Interim" because previously in Sec. 3.1, where ERA-Interim is described, it is said that "Hereafter, it will be referred to as ERA" (P5L30).

10. P7L25: More than 550 "land" cover types.
The word "land" has been added (P7L19).

11. P8L26-27: ...and the third one or final by 8 stations.
The text has been rewritten as "... and the third one by 8 of the 13 stations, which is the final selection" (P8L22-23).

12. P9L13-14: Variability does not mean drought.
In our opinion, the text does not say that variability is drought, but a key aspect of drought analysis, as said in P9L10. Further on in the text (Sec. 4.1) we explain that the resulting values of the standardized indices calculation correspond to the number of standard deviations by which the anomaly deviates from the mean (P10L3-4).

13. P10L12: I am wondering why meteorological drought is represented by SPI-12. There must be a clarification from the authors about the use of SPI-12.
We have extended the text to include the fact that it is a suitable scale considering hydrological resource management (P10L12).

14. P10: In the drought index calculation section, there is no explanation about the SRI, SSMI, and SSI accumulation period. Later I know that it is 1 month.
We propose to include this information in Sec. 4.1 as "On the other hand, the analysis of the RCMs' soil moisture and hydrological drought representation was performed through the calculation of Root Mean Square Difference (RMSD) and the Pearson correlation coefficient (r) at a 95% significance level of the standardized indices computed using an accumulation period of 1 month" (P10L16-17).

15. P11L2: Here you mention SSI-1. You need to mention SRI and SSMI as well.
The text has been modified to include SRI (P11L2). There is no need to include SSMI because it is already mentioned and this paragraph is about the hydrological drought.

16. P11L13: I do not agree that meteorological drought occurs due to an absence of precipitation. Less precipitation can cause drought.
The text has been rewritten as "In this section, we will focus on precipitation because its decrease or absence are the main cause of meteorological drought" (P11L13).

17. P12: Figure 2, from 1989 to 2008. Also for all figures, I prefer the alphabets a, b, c, d, etc are placed in the caption.
The text from the caption has been changed to "from 1989 to 2008" (P12). The captions have been rewritten to place the alphabets a, b, c, d, etc. (P12, P14, and P16).

18. P14: Figure 3. The use of word aggregated is not well fit. Aggregated can have meaning accumulated. It is averaged, right?
The word "aggregated" has been replaced by "averaged" (P14).

19. P16: Figure 5. Actually, the white color in this figure is the area that is not studied.
This is true for Portugal, which is not studied and appears in white. However, some regions across northeastern Spain appear in white on the map showing the correlation between SLR and ERA. Since these regions belong to the study area but are not within the colour scale range, they are represented in white. This has been indicated in the captions of Figs. 5 (P16) and 6 (P18).

20. P16L9: Reference for SPI classification is needed.
The reference is provided in Sec. 4.1 "... according to the SPI drought classification scale (McKee et al., 1993)" (P10L19). In our opinion, it is not necessary to refer to it again. However, we can include it if the referee and editor deem it necessary.

21. P17: Need a color-coding explanation for Table 3 and 4, especially about the gradient colors from red to blue.
The text explaining the color-coding has also been included the caption of Tables 3 (P17), 4 (P19), and 5 (P20).

22. P17L10-12: Please provide the Figure number in the text, thus the reader can easily refer to what Figure. Example: For example, the RS4 and ISB maps show similar patterns (Figure 6a and 6e).
The figure panels have been referenced in the text (P17L10, P18L1,3,5).

23. P18L1: What do you mean with CL4 behavior is quite flat?
We mean that CL4's behaviour is quite constant and homogeneous across the peninsula. The word "flat" has been replaced by "homogeneous" (P18L3).

24. P19: Table 4 caption. Better use SMP instead of SIMPA model. Just to be consistent. I also suggest the author provide the average values of KGE for all station below the last row.
● "SIMPA" has been replaced by "SMP" (P19).
● The average KGE values have been included in Table 4 and referred to in the text (P19L6).

25. P20: For table 5 and 6, I suggest you writing the station names again and writing which column belong to i) and which column belong to ii).
Since the station names are not written in the manuscript, we believe that you propose to include the stations' basins and areas. This has been done. We are sorry but we do not understand the second part of the comment regarding the columns. If you could please explain it again to us, we will see to include the improvement proposed in the manuscript.

26. P21: In the discussion section, why the results are different, are poorly discussed. Example different in the model structures, different in the LSM scheme, or how ET was calculated in the models, or how many subsurface layers the models have, different spatial resolution, etc.
The study analyses how RCMs represent drought and its propagation. In Sec. 6 several issues which can explain the differences found are discussed. The paragraph about LSM (one of the issues treated) has been rewritten (P23L8-13).

27. P22L5: Be careful when you use term intensity for drought. Drought intensity is drought deficit volume/drought duration.
The word "intensity" has been replaced by "severity" in P1L1 and P21L26.

28. P24: Conclusion does not tell the reader about the results. For example which model performs better, how the meteorological drought propagates to hydrological drought, etc.

The model that performs better was explained in Sec. 6. This text has been moved to Sec. 7 and extended (P24L11-18).

We hope you will see a clear improvement in the revised version of the manuscript.

Yours sincerely,

Anaïs Barella-Ortiz

**Relevant changes made in the manuscript**

Most of the changes proposed by the referee have been included in the manuscript. The most relevant ones are:

- Figure and table captions: Color scales have been explained in the captions.
- Table 4: The average KGE has been included.
- Conclusions: The text has been extended to include the model that best performs in each drought type analysed and drought propagation.

[revised manuscript text omitted]